

# AI-based techniques for multi-step streamflow forecasts: Application for multi-objective reservoir operation optimization and performance assessment

Yuxue Guo, Yue-Ping Xu, Xinting Yu, Hao Chen, Haiting Gu, Jingkai Xie

Institute of Hydrology and Water Resources, Civil Engineering and Architecture, Zhejiang University, Hangzhou, 310058, China

*Correspondence to*: *Yue-Ping Xu (yuepingxu@zju.edu.cn)*

**Abstract.** Streamflow forecasts are traditionally effective in mitigating water scarcity and flood defense. This study developed an Artificial Intelligence (AI)-based management methodology that integrated multi-step streamflow forecasts and multi-

objective reservoir operation optimization for water resource allocation. Following the methodology, we aimed to assess forecast quality and forecast-informed reservoir operations performance together due to the influence of inflow forecast uncertainty. Varying combinations of climate and hydrological variables were inputs into three AI-based models, namely Long Short-Term Memory (LSTM), Gated Recurrent Unit (GRU), and Least Squares Support Vector Machine (LSSVM), to forecast short-term streamflow. Based on three deterministic forecasts, the stochastic inflow scenarios were further developed using

Bayesian Model Averaging (BMA) for quantifying uncertainty. The forecasting scheme was further coupled with a multi-reservoir optimization model, and the multi-objective programming was solved using the parameterized Multi-Objective Robust Decision Making (MORDM) approach. The AI-based management framework was applied and demonstrated over a multi-reservoir system (25 reservoirs) in the Zhoushan Islands, China. Three main conclusions were drawn from this study: 1) GRU and LSTM performed equally well on streamflow forecasts, and GRU might be the preferred method over LSTM, given

that it had simpler structures and less modeling time; 2) Higher forecast performance could lead to improved reservoir operation, while uncertain forecasts were more valuable than deterministic forecasts, regarding two performance metrics, i.e., water supply reliability and operating costs; 3) The relationship between forecast horizon and reservoir operation was complex and depended on the operating configurations (forecast quality and uncertainty) and performance measures. This study reinforces the potential of an AI-based stochastic streamflow forecasting scheme to seek robust strategies under uncertainty.

**Keywords:** AI-based techniques, multi-step streamflow forecasts, reservoir operation, forecast quality, forecast horizon.

## 1 Introduction

Multi-step streamflow forecast is of great importance for reservoir operations to determine optimal water allocation considering the current use and the carry-out storage for mitigating water scarcity risk in the future (Guo et al., 2018; Zhao et

al., 2019). Previous studies have identified that real-time reservoir operations are influenced by multiple uncertainties (Xu et





al., 2020), among which inflow forecast uncertainty has been determined as the primary source, resulting in the risk of water shortage when the forecast inflow overestimates the actual inflow. Ensemble forecasting techniques are commonly used to characterize various uncertainties in streamflow forecasts. According to comparative analysis for various probabilistic forecasting techniques (Nott et al., 2012; Fang et al., 2018a; Zhai and Chen, 2018; Zhou et al., 2020b), Bayesian Model

Averaging (BMA) (Hoeting et al., 1999) has been found to be an effective and most commonly used method to evaluate uncertainty and thus can be used in streamflow forecast.

Any ensemble forecast approach relies upon model diversity that different models produce, with specific emphasis and different aspects of the features they want to model (Zhou et al., 2020a). In the last few decades, many approaches have been developed to forecast streamflow, including physically-based and data-driven models (Tikhamarine et al., 2020; Zuo et al.,

2020). Although physically-based models can help understand underly physical processes, they usually require a large amount of input information, such as meteorological data, geographic data, soil, and land use characteristics (Guo et al., 2018; Guo et al., 2020a). Different from physically-based models, data-driven models based on statistical modeling have attracted significant interests due to their simplicity and satisfactory forecast results with low information requirements (Al-Sudani et al., 2019; Mehdizadeh et al., 2019; Osman et al., 2020). Artificial intelligence (AI)-based approaches, i.e., machine learning (ML)

methods, belong to the latter group. The widely used ML approaches include Artificial neural network (ANN) and Least Squares Support Vector Machine (LSSVM) (Ghumman et al., 2018; Kisi et al., 2019; Meng et al., 2019; Adnan et al., 2020; Ali and Shahbaz, 2020). Such models have been proven to be efficient tools to model qualitative and quantitative hydrological variables and deal with non-linear features in streamflow. In recent years, the booming development of deep learning technology has brought many new approaches, such as recurrent neural networks (RNNs) (Elman, 1990), one of the most

popular neural networks in the deep learning field. RNNs can preserve and remember the short-past and long-past information and thus are preferred for a complex and highly non-linear timing problem. Long Short-Term Memory (LSTM) (Hochreiter and Schmidhuber, 1997) and Gated Recurrent Unit (GRU) (Cho et al., 2014) are two different versions of RNNs. They have been demonstrated to generate comparable performances, but GRU has a more straightforward structure and a higher operation speed than LSTM. Although LSTM and GRU networks have been successfully applied in many fields (Greff et al., 2017;

Zhang et al., 2018; Gao et al., 2020; Jung et al., 2020), few applications that assessed them together have been found in the hydrological field.

Nevertheless, while a considerable research effort has been made to evaluate and improve the quality of streamflow forecasts (Gibbs et al., 2018; Nanda et al., 2019; Sharma et al., 2019; Van Osnabrugge et al., 2019; Feng et al., 2020; Pechlivanidis et al., 2020), how forecasts impact decision-making in the real-time reservoir operations has not been investigated thoroughly,

e.g., do high-quality forecasts mean improved decision? Traditionally, a skillful forecast is vital for the reliability of the forecasts and is essential to promote the use of forecasts in real-world applications by decision makers. In fact, forecast value is expected to increase with forecast quality, but it may also vary based on other factors such as reservoir capacity and operating objectives (Anghileri et al., 2016). Some studies even have disproved the intuitive assumption that higher forecast performance always leads to better operation decisions, for example, agricultural water management (Chiew et al., 2003) and water





resources allocation (Turner et al., 2017). Therefore, when forecasts are used to support reservoir operation, they should be assessed in which conditions they can help make better decisions. Moreover, forecast uncertainty and error generally grow up with the increase of the forecast horizon (Maurer and Lettenmaier, 2004; Denaro et al., 2017; Zhao et al., 2019). A decision maker may doubt whether longer forecast lead times provide more sufficient information for a decision purpose or not. It is crucial to determine an efficient forecast lead time that can provide appropriate inflow information for reliable reservoir release

decisions for making the best use of forecast information. However, few studies have demonstrated the applicability and effectiveness of the forecast horizon in a forecast-based reservoir operation system (Xu et al., 2014; Anghileri et al., 2016). There is a continuous need for in-depth study to conduct posterior evaluations of forecasts with different forecast lead times and obtain the efficient forecast horizon for water allocation.

A decision maker must allocate limited water to different water use sectors considering the conflicting objectives (e.g., benefits

and costs) and multiple uncertainties (e.g., forecast uncertainty) in a forecast-based reservoir operation system. Multi-objective programming (MOP) is a useful tool for helping decision makers facilitate decision making with multiple conflicting objectives (Fang et al., 2018b; Guo et al., 2020c), which can offer feasible methods for generating compromise decision alternatives. Some MOP approaches have been widely developed to tackle the uncertainty associated with the decision making processes, such as multi-objective fuzzy programming (Zimmermann, 1978; Pishvaee and Razmi, 2012; Ren et al., 2017) and multi-

objective stochastic programming (Xu et al., 2014; Xu et al., 2020; Zhang et al., 2020). These approaches generally convert the multi-objective functions into a single-objective deterministic problem through a fuzzy programming method or a constraint operator. They can effectively deal with the uncertainties between objectives and/or constraints by integrating the decision makers' aspiration levels. However, they may encounter difficulties due to the need for pre-determined individual preference or reasonable bounds for all objectives. In comparison, multi-objective robust decision making (MORDM) is an

effective way to handle such difficulties (Kasprzyk et al., 2013; Yan et al., 2017). It can generate many alternative solutions (Pareto solutions) that do not require assumptions about decision makers' preferences and enhance the robustness of the optimization process. Besides, MORDM, by parameterizing the decision space, can avoid the curse of dimensionality in some MOP approaches, and simplify computational complexity and reduce the running time (Giuliani et al., 2016; Salazar et al., 2017).

In summary, there are still several challenges in forecast-informed reservoir optimization. To address these challenges, the specific research questions of this study are:

(1) Can GRU achieve the same accuracy in the streamflow forecast compared to LSTM with fewer parameters and more straightforward structures?

(2) In which conditions can an improvement in forecast skill be translated into an improvement in reservoir operation

optimization?

(3) How do such short-term inflow forecasts with different forecast horizons be used to optimize the multi-reservoir system to impact operation results?





To answer the questions mentioned above, we build an AI-based management framework, which integrates multi-step streamflow forecasts and multi-reservoir operation optimization. We strive to: (1) simulate inflow using LSTM, GRU, and LSSVM and verify their effectiveness on short-term deterministic streamflow forecasts; (2) generate stochastic inflow scenarios using BMA for refining uncertainty characterization; (3) develop the parameterized MORDM framework for a multi-reservoir operation system and inform decision making by assessing the value, that is, the operation benefits gain or the induced cost of forecasts with a particular lead time. As a case study, including one recipient reservoir storing water from the continental diversion project and 24 supply reservoirs storing water from local rainfall, 25 reservoirs supplying water for four water plants in the Zhoushan Islands, China, are chosen to assess the performance of the AI-based forecast and the forecast-informed operation.

## 2. Methodology

The experimental approach followed in the study is shown in Figure 1 and described in the following sections.

*Figure 1 is here*

### 2.1 Machine learning (ML) methods

This part gives a brief introduction to long short-term memory (LSTM), gated recurrent unit (GRU), and least square support vector machine (LSSVM).

#### 2.1.1. Long short-term memory (LSTM)

LSTM network is one of the recurrent neural networks (RNNs) developed by Hochreiter and Schmidhuber (1997), and the basic structure of an LSTM cell is illustrated in Figure 2(a). It is an improved RNN aiming to solve problems such as gradients in long-term memory and backpropagation. The LSTM cell has three gates maintaining and adjusting its cell state and hidden state, including the forget gate, input gate, and output gate. The forget gate determines what information would be thrown away from the cell state. The input gate decides which information is used to update the cell state. The output gate controls which information stored in the current cell state flows into the new hidden state. In Figure 2(a), the state ($c_t$), and the hidden state ($h_t$) of the LSTM cell are updated as follows (Hochreiter and Schmidhuber, 1997):

Forget gate: $\qquad f_t = \sigma(W_f x_t + U_f h_{t-1} + b_f)$, $\qquad\qquad$ (1)

Input gate: $\qquad i_t = \sigma(W_i x_t + U_i h_{t-1} + b_i)$, $\qquad\qquad$ (2)

Potential cell state: $\qquad \tilde{c}_t = tanh\left(W_{\tilde{c}} x_t + U_{\tilde{c}} h_{t-1} + b_{\tilde{c}}\right)$, $\qquad\qquad$ (3)

Cell state: $\qquad c_t = f_t \odot c_{t-1} + i_t \odot \tilde{c}_t$, $\qquad\qquad$ (4)

Output state: $\qquad o_t = \sigma(W_o x_t + U_o h_{t-1} + b_o)$, $\qquad\qquad$ (5)


Hidden state: $\qquad h_t = o_t tanh \odot c_t,$ (6)

where $f_t$, $c_t$, $o_t$ and $\tilde{c}_t$ represent the forget gate, input gate, output gate, and potential cell state, respectively. $\odot$ denotes the element-wise multiplication of vectors, $tanh(\cdot)$ is the hyperbolic tangent; $x_t$ represents the current input vector, $h_{t-1}$ denotes the last hidden cell state and the initial state of $h_t$ is $h_0 = 0$. $\sigma(\cdot)$ represents the logistic sigmoid function. $[W_f, W_i, W_o, W_{\tilde{c}}]$, $[U_f, U_i, U_o, U_{\tilde{c}}]$, and $[b_f, b_i, b_o, b_{\tilde{c}}]$ represent the input weight matrix, recurrent weight matrix, and bias vectors for the

forget, input-output, and potential cell gates, respectively.

### 2.1.2. Gated recurrent unit (GRU)

GRU networks were proposed as a modification of LSTM networks with a more straightforward structure (Cho et al., 2014). The specific structure of the GRU cell is shown in Figure 2(b). Compared with LSTM, GRU has only two control gates, including a reset gate and an update gate. The update gate is applied to control how much information of the previous step is

brought into the current step, while the reset gate is used to control the degree of ignoring the information of the previous state. In this way, GRU is superior to LSTM in terms of computer modelling time and parameter updates. In Figure 2(b), the state ($c_t$) and the hidden state ($h_t$) of the GRU cell are updated as follows (Cho et al., 2014):

Reset state: $\qquad r_t = \sigma\left(W_r x_t + U_r h_{t-1} + b_r\right),$ (7)

Update state: $\qquad z_t = \sigma\left(W_z x_t + U_z h_{t-1} + b_z\right),$ (8)

Potential cell state: $\qquad \tilde{c}_t = tanh\left(W_{\tilde{c}} x_t + U_{\tilde{c}}\left(r_t \odot h_{t-1}\right) + b_{\tilde{c}}\right),$ (9)

Cell state: $\qquad c_t = \left(1 - z_t\right) \odot c_{t-1} + z_t \odot \tilde{c}_t,$ (10)

Hidden state: $\qquad h_t = c_t,$ (11)

where $r_t$, $z_t$, and $\tilde{c}_t$ represent the reset, update, and potential cell state, respectively. $\odot$ denotes the element-wise multiplication of vectors, $tanh(\cdot)$ is the hyperbolic tangent; $x_t$ represents the input vectors, $h_{t-1}$ denotes the last hidden cell state

and the initial state of $h_t$ is $h_0 = 0$. $\sigma(\cdot)$ represents the logistic sigmoid function. $[W_r, W_z, W_{\tilde{c}}]$, $[U_r, U_z, U_{\tilde{c}}]$, and $[b_r, b_z, b_{\tilde{c}}]$ represent the input weight matrix, recurrent weight matrix, and bias vectors for the reset, update, and potential cell gates, respectively.

*Figure 2 is here*





### 2.1.3. Least squares support vector machine with grey wolf optimizer (GWO-LSSVM)

LSSVM is a modified version of SVM, proposed by Suykens and Vandewalle (1999), to reduce the computational time of SVM. SVM uses the quadratic program to formulate the training process of modeling procedure, while LSSVM aims to employ the least-squares loss functions. The LSSVM non-linear function is expressed as (Suykens et al., 2002):

$$f(x) = w^T \varphi(x) + b , \tag{12}$$

where $\varphi(\cdot)$ is the mapping function that maps the input $x$ into a d-dimensional feature vector, $w$ is a weight vector, and $b$ represents bias. In LSSVM, a minimum objective function is proposed to estimate $\omega$ and $b$ (Suykens et al., 2002).

$$\min J(w,e) = \frac{1}{2} w^T w + \frac{1}{2} \gamma \sum_{i=1}^{N} e_i^2 , \tag{13}$$

that has the following constraints (Suykens et al., 2002):

$$y_i = w^T \varphi(x_i) + b + e_i , \tag{14}$$

where $e$ is the error variable and $\gamma$ is the regulative constant. The objective function can be obtained to solve the optimization problems in Eq. (13) by introducing the Lagrange multipliers $\alpha$ and transferring the constraint problem into an unconstrained one (Suykens et al., 2002):

$$L(w,b,e,\alpha) = \frac{1}{2} w^T w + \frac{1}{2} \gamma \sum_{i=1}^{N} e_i^2 - \sum_{i=1}^{N} \alpha_i \left( w^T \varphi(x_i) + b + e_i - y_i \right) , \tag{15}$$

By finding the partial derivative of Eq. (15) with respect to $w$, $b$, $\alpha_i$, and $e_i$, the following equation can be derived:

$$y = \sum_{i=1}^{N} \alpha_i \left( \varphi(x)^T \varphi(x_i) \right) + b = \sum_{i=1}^{N} \alpha_i K(x, x_i) + b , \tag{16}$$

where $K(x, x_i)$ is the kernel function. Many kernel functions such as linear, polynomial, radial basis, and sigmoidal have been proposed for LSSVM (Bemani et al., 2020). We adopt the most widely used kernel function, Radial Basis Function (RBF), in this study. The RBF is expressed as:

$$K(x, x_i) = \exp\left( -\|x - x_i\|^2 / 2\sigma^2 \right), \tag{17}$$

where $\sigma^2$ is the kernel parameter. In this study, the parameter $\gamma$ and $\sigma$ were optimized using grey wolf optimizer (GWO). Please see more details on GWO in Guo et al. (2020d).

### 2.2 Bayesian model averaging (BMA)

Generally, it is difficult to determine which model is the best one, leading to model uncertainty. BMA is proposed to solve the uncertainty of the model through averaged estimations from individual models (Liu and Merwade, 2019; Samadi et al., 2020). The weight for each model is based on the simulated decision probability density function, i.e., the posterior probability of the





model $f_k$. Suppose $Q$ is the unknown quantity we want to predict, given a subset of forecast models $f = \{f_1, f_2, \ldots, f_K\}$

( $k = 1, 2, \ldots, K$ ) and the observed data $D$, the posterior distribution of $Q$ can be calculated as (Hoeting et al., 1999):

$$p(Q|D) = \sum_{i=1}^{K} p(f_k|D) \cdot p_k(Q|f_k, D) = \sum_{i=1}^{K} w_k \cdot p_k(Q|f_k, D), \tag{18}$$

where $p_k(Q|f_k, D)$ is the posterior distribution of $Q$ given the model $f_k$ and the observed data $D$, and $p(f_k|D)$ is the

posterior probability. In this case, posterior probabilities are the weighting factor for each model, and $\sum_{k=1}^{K} w_k = 1$. The posterior

mean (E) and variance (V) of $Q$ are as follows (Hoeting et al., 1999):

$$E[Q|D] = \sum_{i=1}^{K} w_k \cdot E[p_k(Q|f_k, D)] = \sum_{i=1}^{K} w_k f_k, \tag{19}$$

$$V[Q|D] = \sum_{i=1}^{K} w_k \cdot \left[ f_k - \sum_{i=1}^{K} w_k f_k \right]^2 + \sum_{i=1}^{K} w_k \sigma_k^2, \tag{20}$$

where $\sigma_k^2$ is the variance of the model $f_k$. BMA weights can be calculated using optimization methods. In this study, the

Expectation-Maximization (EM) is used to identify the BMA parameters (weight $w_k$ and variance $\sigma_k^2$) and then to estimate

the release interval (Lee et al., 2020). Details of BMA can also be found in Hoeting et al. (1999).

**2.3 Forecast performance measures**

Three performance indicators were applied to assess the deterministic forecast performance of the three data-process models.

They were Nash-Sutcliffe efficiency (NSE) (Nash and Sutcliffe, 1970), root mean square errors (RMSE) (Karunanithi et al.,

1994), and mean absolute error (MAE) (Legates and McCabe Jr., 1999). They are expressed as below.

$$NSE = 1 - \frac{\sum_{t=1}^{T} (Q_{m,t} - Q_{o,t})^2}{\sum_{t=1}^{T} (Q_{o,t} - \bar{Q}_o)^2}, \tag{21}$$

$$RMSE = \sqrt{\frac{1}{T} \sum_{t=1}^{T} (Q_{m,t} - Q_{o,t})^2}, \tag{22}$$

$$MAE = \frac{1}{T} \sum_{t=1}^{T} |Q_{m,t} - Q_{o,t}|, \tag{23}$$

where $T$ is the number of samples; $Q_{m,t}$ is the forecasted reservoir inflow (m³/s); $Q_{o,t}$ is the observed inflow (m³/s), and $\bar{Q}_o$

is the average of the observed inflow (m³/s). The NSE can be used to evaluate the stability of the forecasted value. In contrast,

RMSE and MAE are used to characterize the overall forecast accuracy. NSE value is (−∞, 1], while MAE and RMSE values





are $(0, +\infty)$. Generally, models with larger values of NSE or smaller values of RMSE and MAE provide better forecasting accuracy.

In addition, two performance indicators were used to evaluate the performance of ensemble forecast models, i.e., the containing ratio (*CR*), and average deviation amplitude (*D*), were adopted for assessing the goodness of the prediction bounds (Xiong et al., 2009).

$$CR = \frac{1}{T}\sum_{t=1}^{T} N_t \times 100\% \quad N_t = \begin{cases} 1 & \text{if } \hat{Q}_{l,t} \le Q_{o,t} \le \hat{Q}_{u,t} \\ 0 & \text{else} \end{cases}, \tag{24}$$

$$D = \frac{1}{T}\sum_{t=1}^{T} \left| \frac{1}{2}\left(\hat{Q}_{l,t} + \hat{Q}_{u,t}\right) - Q_{o,t} \right|, \tag{25}$$

where $\hat{Q}_{l,t}$ and $\hat{Q}_{u,t}$ represent the lower and upper prediction bounds of streamflow (m$^3$/s), respectively. Clearly, models with higher *CR* values but lower *D* values would produce better performance.

**2.4 Parameterized multi-objective robust decision making (MORDM)**

Multi-objective robust decision making (MORDM) provides a mechanism for stress-testing operational alternatives under uncertainty. We identify and evaluate different operational transfer strategies for water allocation in the Zhoushan Islands using the MORDM method. The main steps of the MORDM framework are (Hadjimichael et al., 2020): (1) problem formulation, including the possible actions (i.e., decision variables) and performance measures; (2) generating alternative management actions using multi-objective evolutionary algorithms (MOEAs); (3) perform an uncertainty analysis and identify robust solutions. Problem formulation is a critical step in the MORDM framework (Zeff et al., 2014). To reinforce reservoir operation under uncertain forecasts, the objectives are instead evaluated over stochastic inflows. The uncertainties are then mitigated using a robust approach (Giuliani and Castelletti, 2016; Guo et al., 2020b), e.g., the principle of insufficient reason, minimax, and minimax regret approaches.

In general, the decision variables in the multi-reservoir optimization problem are the volumes of water to be allocated each day. Open-loop strategies prefer each decision in a time series as an independent decision. In contrast, closed-loop control strategies prescribe decisions conditioned on system state variables (Quinn et al., 2017a). We use the direct policy search (DPS), where the rules for operational strategies are approximated as non-linear functions that vary with specific parameters and system states to derive closed-loop control strategies (Giuliani et al., 2016; Quinn et al., 2017b; Salazar et al., 2017). DPS is based on the parameterization of the operating policy $p_\theta$ and the exploration of the parameter space $\Theta$ to find a parameterized policy that optimizes the expected long-term cost, i.e.,

$$p_\theta^* = \arg\min_{p_\theta} J_{p_\theta} \quad s.t. \ \theta \in \Theta, \tag{26}$$





where $J$ is the objective function. $p_\theta^*$ is the corresponding optimal policy with parameters $\theta^*$. Different DPS approaches

have been proposed (Deisenroth et al., 2013). In this study, we use Radial Basis Functions (RBFs) to parameterize the policy

and the $k^{\text{th}}$ decision variable in the vector $u_t$ (with $k = 1, \ldots, K$ ) is defined as:

$$u_t^k = \sum_{i=1}^{N} \omega_{i,k} \varphi_{i,k}(\Gamma_t) , \qquad (27)$$

where $N$ is the number of RBFs $\varphi(\cdot)$, and $\omega_{i,k}$ is the weight of the $i^{\text{th}}$ RBF, $\sum_{i=1}^{N} \omega_{i,k} = 1 \quad \omega_{i,k} > 0$. The single RBF is defined as

follows:

$$\varphi_{i,k}(\Gamma_t) = \exp\left[ -\sum_{j=1}^{M} \frac{\left[ (\Gamma_t)_j - c_{j,i} \right]^2}{b_{j,i}^2} \right] , \qquad (28)$$

where $M$ denotes the number of policy inputs $\Gamma_t$ and $c_i$, $b_i$ are the M-dimensional center and radius vectors of the $i^{\text{th}}$ RBF,

respectively. The centers of the RBF must lie within the bounded input space (Yang et al., 2017). The parameter vector θ is

defined as $\theta = \left[ c_{i,j,k}, b_{i,j,k}, \omega_{i,j,k} \right]$ with the number of $\theta$ is $n_\theta = N \times K \times (2 \times M + 1)$.

The parameterized MORDM approach is then coupled with a rolling horizon scheme to solve the short-term reservoir operation

problem. Given the lead time of 7 days as an example, it is operated following two steps: the optimization model is first

operated daily over a 7-day horizon using the parameterized MORDM; after implementing current water allocation decisions,

the status, inflow, and other information of reservoirs update as time evolves, and then the remainder is subsequently operated.

The two steps are repeated until the process is completed.

## 3. Case study

### 3.1 Study area and data

The Zhoushan Islands are located in the northeast of Zhejiang Province, China, with a total area of 22,000 km2 and 1,390

islands (Figure 3). The climate is governed by monsoon-influenced subtropical marine weather systems, and the annual mean

temperature and precipitation are 17 ℃ and 1,300 mm, respectively. There are no large rivers in the islands, and the insufficient

freshwater resources severely limit the development of industry and population in Zhoushan. Recently, a continental diversion

project transferring water from Ningbo City to Zhoushan is treated as an effective solution to partially overcome the water

scarcity problem. The transferred water is stored in Huangjinwan Reservoir and then operated together with the limited

freshwater resources in the remaining 24 reservoirs to supply water to four water plants, i.e., Daobei, Hongqiao, Lincheng, and

Pingyangpu. Data for this study includes historical inflow and state of reservoirs, water demand of water plants, and climate



forcing data over 2002-2008. The climate data, including daily precipitation and evaporation, are observed at one
meteorological station and three rainfall stations. The characteristics of the reservoirs are listed in Table 1.

*Figure 3 is here*

*Table 1 is here*

**3.2 Problem formulation**

The main goal of the water allocation plan is to ensure sufficient water flows into the four plants in the Zhoushan Islands. This
is realized by allocating water in Huangjinwan Reservoir and the remaining 24 reservoirs. Figure 4 shows the simplified
schematic diagram of the operating system for the functions of water supply. According to the water pipe flow direction, the
whole islands are divided into three districts, i.e., Daobei, Hongqiao, and Dongbu.

*Figure 4 is here*

Three objectives are identified to evaluate the performance of the strategies. The conflicting objectives are to minimize the
water deficiency ratio of the Daobei Plant, minimize the water deficiency ratio of the remaining three plants (Hongqiao,
Lincheng, and Pingyangpu), and maximize the net benefits. The three plants can feed each other and thus are considered
together in our study. A decision-maker would consider a different suite of costs depending on whether an existing system is
being managed or a completely new system is being designed. As water supply occurs in an existing system, costs considered
in this study are the operating costs. These objective functions are given as follows:

$$\text{Min} \quad f_1(x) = \left( \sum_{t=1}^{T} W_{db,n,t} - \sum_{t=1}^{T} W_{db,s,t} \right) \bigg/ \sum_{t=1}^{T} W_{db,n,t} \times 100\% \; , \tag{29}$$

$$\text{Min} \quad f_2(x) = \left( \sum_{i=1}^{3}\sum_{t=1}^{T} W_{n,t,i}^{th} - \sum_{i=1}^{3}\sum_{t=1}^{T} W_{s,t,i}^{th} \right) \bigg/ \sum_{i=1}^{3}\sum_{t=1}^{T} W_{n,t,i}^{th} \times 100\% \; , \tag{30}$$

$$\text{Min} \quad f_3(x) = \left( M_{c,\text{island}} + M_{c,\text{mainland}} \right) - M_r \; , \tag{31}$$

where $f_1$ and $f_2$ are the water deficiency ratio of Daobei Plant and the sum of the remaining three plants, respectively (%); $f_3$ is
the net operating costs (RMB); $W_{db,n,t}$ and $W_{db,s,t}$ are the amount of water supply, and demand for Daobei Plant, respectively
(m³); $W_{n,t,i}^{th}$, and $W_{s,t,i}^{th}$ are the amounts of water supply and demand for the remaining three plants, respectively (m³); $M_{c,\text{island}}$
and $M_{c,\text{mainland}}$ are the costs for water supply from the islands and the mainland, respectively (RMB); $M_r$ is the revenue (RMB).
The revenue can be obtained according to:

1) Operating costs for water supply from islands ($M_{c,\text{island}}$, RMB)

$$M_{c,island} = M_{c,island,1} + M_{c,island,2} + M_{c,island,3} \; , \tag{32}$$

$$M_{c,island,1} = c_{island,1} \times \sum_{j=1}^{J}\sum_{t=1}^{T} W_{s,t,j} = c_{island,1} \times \sum_{j=1}^{J}\sum_{i=1}^{I}\sum_{t=1}^{T} Q_{i,t} \Delta t \; , \tag{33}$$



$$M_{c,island,2} = c_{island,2} \times \sum_{j=1}^{J} \sum_{t=1}^{T} W_{s,t,j} = c_{island,2} \times \sum_{j=1}^{J} \sum_{i=1}^{I} \sum_{t=1}^{T} Q_{i,t} \Delta t , \tag{34}$$

$$M_{c,island,3} = c_{island,3} \times \sum Q_{island} / \left( \sum_{n=1}^{N} Q_{max\_n} \times 3600s / \sum_{n=1}^{N} P_n / 1h \right), \tag{35}$$

where $M_{c,island,1}$, $M_{c,island,2}$, and $M_{c,island,3}$ represent the water resource fees paid to the government, water fees paid to reservoir managers, and the electricity fees in Zhoushan City, respectively (RMB); $c_{island,1}$, $c_{island,2}$, and $c_{island,3}$ denote the constant vectors,

representing the unit price of water resources, the unit price of water, and the electric unit price in Zhoushan, respectively (RMB/m³); $Q_{island}$ denotes the amount of water flowing through the pumping station (m³/s); $N$ denotes the numbers of a pumping set; $P_n$ denotes the supporting motor power of the $n^{th}$ pump (Kw), and $Q_{max\_n}$ denotes the upper flow boundary of the $n^{th}$ pump (m³/s).

2) Operating costs for water supply from the mainland ($M_{c,mainland}$, RMB)

$$M_{c,mainland} = M_{c,mainland,1} + M_{c,mainland,2} + M_{c,mainland,3} , \tag{36}$$

$$M_{c,mainland,1} = c_{mainland,1} \times \sum_{t=1}^{T} x_{mainland,t} \Delta t , \tag{37}$$

$$M_{c,mainland,2} = c_{mainland,2} \times \sum_{t=1}^{T} x_{mainland,t} \Delta t , \tag{38}$$

$$M_{c,mainland,3} = c_{island,3} \times \sum_{l=1}^{3} \sum_{t=1}^{T} \times \left( (L + Q_{mainland}/S)/Q_{max}/3600/S \right)/3600 , \tag{39}$$

where $M_{c,mainland,1}$, $M_{c,mainland,2}$, and $M_{c,mainland,3}$ represent the water resources fees paid to the government, water fees paid to the river managers, and electricity fees in Ningbo City, respectively (RMB); $c_{mainland,1}$, $c_{mainland,2}$, and $c_{mainland,3}$ denote the constant vectors, representing the unit price of water resources, the unit price of mainland water, and the electric unit price in Ningbo, respectively (RMB/m³); $Q_{mainland}$ denotes the amount of water transferred from Ningbo (m³); $L$ denotes the total length of the continental diversion pipeline (m); $S$ denotes the cross-sectional area of the continental diversion pipeline (m²), and $Q_{max}$

denotes the upper flow boundary (m³/s).

3) Revenues ($M_r$, RMB)

$$M_r = b \times \left( \sum_{s=1}^{S} \sum_{t=1}^{T} \sum_{i=1}^{t} W_{s,t,i} \right), \tag{40}$$

where $b$ denotes the unit price of water supply revenue (RMB/m³), and $W_{s,t,i}$ is the amount of water that a reservoir supplies to a waterworks at a given time (m³).

The optimization model is subject to the following constraints:

(1) Water balance: $\quad\quad\quad V_{t+1,i} = V_{t,i} + \left( I_{t,i} - Q_{t,i} \right) \Delta t , \tag{41}$

(2) Reservoir storage: $\quad\quad\quad V_{min,i} \leq V_i \leq V_{max,i} , \tag{42}$





(3) Release capacity limits:  $Q_i \leq Q_{max,i}$ ,                                                                                      (43)

(4) Pumping station limits:  $Q_{t,i}^p \leq Q_{max,t,i}^p$ ,                                                                                (44)

where $\Delta t$ is the time step; $i$ is the number of a reservoir; $I$ and $Q$ are the inflow and release, respectively (m³/s); $V$ is the reservoir storage (m³); $V_{min}$ and $V_{max}$ are the lower and upper storage boundaries, respectively (m³); $Q_{max}$ is the maximum reservoir release (m³/s); $Q^p = \sum_{j=1}^{n} Q_j$  represents the water pumped by the pumping station (m³/s), and $Q_{max}^p$  is the maximum pumping capacity (m³/s).

**3.3 Forecast inputs setting**

In this study, five input combination scenarios were considered to investigate whether the use of data-driven methods with climate forcing is efficient in inflow forecasts or not. These scenarios are described in Table 2. $P_a$ represents antecedent precipitation, $E_a$ represents antecedent evaporation, $Q_a$ represents antecedent streamflow, $P_f$ represents forecast precipitation, and $E_f$ represents forecast evaporation.

*Table 2 is here*

Several strategies have been proposed in the literature to tackle a multi-step-ahead forecast task (Kline, 2004), such as the recursive, direct, combination of direct and recursive strategies. In this study, we chose one of the most carried out strategies, i.e., the direct strategy (Ben Taieb et al., 2012), to forecast multi-step streamflow over the short-term horizon (1-7 days). In this case, the streamflow is forecasted using the following equations, given S3 as an example.

$$1d: Q_{t+1}^f = F\left(Q_t, Q_{t-1}, ..., Q_{t-k}, E_t, E_{t-1}, ..., E_{t-k}, P_{t-1}, ..., P_{t-k}\right)$$
$$2d: Q_{t+2}^f = F\left(Q_t, Q_{t-1}, ..., Q_{t-k}, E_t, E_{t-1}, ..., E_{t-k}, P_{t-1}, ..., P_{t-k}\right)$$
$$...$$
$$7d: Q_{t+7}^f = F\left(Q_t, Q_{t-1}, ..., Q_{t-k}, E_t, E_{t-1}, ..., E_{t-k}, P_{t-1}, ..., P_{t-k}\right)$$

where $F()$ is the mapping function between inputs and outputs.

**4. Results and discussion**

**4.1 Multi-step deterministic forecasts based on ML methods**

An issue with the ML methods is that they can easily overfit training data. To avoid this issue, the entire data is divided into three subsets in RNNs: (i) a training set, which is used to compute the gradient and update the weights and biases of the

network, (ii) a validation set over which the errors are monitored during the training process and is used to decide when to stop training, (iii) a test set, which is used to assess the expected performance in the future. In addition, dropout is a regularization method where input and recurrent connections to LSTM and GRU units are probabilistically excluded from activation and weight updates while training a network. The strategies mentioned above have the effect of reducing overfitting and improving





model performance in RNNs. As for LSSVM, we avoid overfitting by minimizing the NSE during the calibration, validation, and test periods. In this study, Jan 2002 to Dec 2006 was used as the training period, while the validation and tests extended from Jan 2007 – Dec 2007 and Jan 2008 - Dec 2008, respectively. The NSE was calculated for each lead-time of the modeled flow for assessing the performance.

We considered the five different input scenarios described in Section 3.3. Table 3 demonstrates the forecast analysis carried out with the different configurations (input combination and forecast model), tabulating the NSE ranges for lead times from 1 day-ahead to 7 day-ahead over all reservoirs during the calibration, validation, and test periods. It can be seen that S1 using only the flow variables and S2 using only the antecedent climate variables are inferior to the other scenarios. The performance is generally improved when the flow variables are used in combination with the antecedent precipitation and evaporation under S3. However, in this case, the antecedent variables succeed to forecast only at 1-day ahead. The forecast performance decreases significantly as the forecast horizon increases from 1-day to7-day ahead. Herein, we suppose that the following precipitation and evaporation have been forecasted. It is clear that S4 and S5, with the forecast climate variables, make significant increments in streamflow forecasting. The NSE can remain relatively stable at different horizons. There are no apparent differences between the three forecast models during the calibration period. However, the two RNNs perform better than GWO-LSSVM during the validation period, while GWO-LSSVM outperforms during the test periods. Besides, given that GRU has more superficial structures and fewer parameters and requires less time for model training, it may be the preferred method for short-term streamflow forecast compared with LSTM. Same results have been obtained in Gao et al. (2020) when they used LSTM and GRU to model short-term rainfall-runoff relationships.

*Table 3 is here*

We aimed to compare how the forecasted climate variables impact the streamflow forecast and reservoir operation performance. For the sake of brevity, S3 and S5 were compared in detail in the following section. Recall that S3 uses flow variables, antecedent precipitation, and evaporation as inputs, while S5 uses flow variables as well as the antecedent and forecast climate forcing. After assessing model validity, the next step was to compare the performance across the 24 reservoirs. The coefficient of variation (COV), defined as the ratio of the standard deviation of the inflow time series, was used to capture the varying characteristics of the incoming flow into the reservoir. From Figure 5, it reveals a strong negative relationship between COV and forecast performance under S3 at all lead times. The forecast performance decreases as the COV increases for all forecast models. This indicates that the more variation the flow has, the harder it is for data-driven methods to learn the flow pattern when there exists not enough input information. However, the negative signal under S5 (Figure 6) with forecasted climate variables (precipitation and evaporation in this study) is not that strong as it under S3, which indicates again that the forecast climate variables can help AI-based models mapping functions between inputs and outputs. The improvements are more significant for the two RNN models, i.e., LSTM and GRU. This result demonstrates that the efficiency of deep-learning RNN methods is better and more accurate than LSSVM.

*Figure 5 is here*


*Figure 6 is here*

**4.2 Multi-step stochastic forecasts based on BMA method**

Based on the forecast results of three data-driven models in the calibration period, the BMA method determined weights for

LSTM, GRU, and GWO-LSSVM models. The weights reflecting the performance of the ensemble models during the calibration period are shown only for lead times of 1 and 7 days for the sake of brevity under S3 and S5 in Figure 7. The model weights reflect the comparative importance of all the competitive modelling predictions on one level. Figure 7 indicates that it is difficult to conclude which individual model provides the best prediction. For example, GRU outperforms the remaining two models for Hongqiao Reservoir, while LSTM performs best for Cenggang Reservoir in Figure 7(a1). Similar results can

be obtained from Figure 7(b1). Comparatively, Figure 7(a2) shows that LSTM and GWO-LSSVM influence the BMA model more than GRU. This higher weight is assigned because the forecasts are more similar to observations than those less similar to observations using the BMA posterior processor. However, observed from Figure 7(b2), the prediction accuracy of GWO-LSSVM is seriously affected, and much less than that of GRU. It is consistent with the results obtained in Figure 6, indicating that RNNs outperform GWO-LSSVM when there exists more input information under S5. Overall, model uncertainty always

exists whether forecast climate variables are involved or not, and it is necessary to analyze and evaluate the model uncertainty involved using BMA.

*Figure 7 is here*

To access model validity, the evaluation of the modeled streamflow is performed over calibration, validation, and test periods using NSE, RMSE, and MAE metrics. Table 4 shows the performance metric ranges for all 24 reservoirs of BMA methods

under S3 and S5. Apparently, both the replicative (forecast performance in calibration sets) and predictive (forecast performance in validation and test sets) validity under S5 for forecast horizons are significantly better than those under S3. For example, Figure 8 demonstrates the improvement rates in terms of NSE, RMSE, and MAE of the BMA model compared with the three individual models. BMA produces the maximum NSE, minimum RMSE, and minimum MAE during the calibration period for both two scenarios, indicating that BMA has the best goodness-of-fit. This is because the weights are derived

according to the individual forecast model in this period. With respect to validation and test periods, the BMA method shows better forecasts than the three comparative models except for the GRU modeling validation datasets under S5. Thus, it is shown that the BMA model well matches the actual streamflow.

*Table 4 is here*

*Figure 8 is here*

The model validity was then assessed using (i) hydrographs and (ii) scatter plots of observed and modeled streamflow, as shown in Figure 9 and Figure 10. Herein, we only show three reservoirs, i.e., Hongqiao (the largest reservoir), Goushan (the medium reservoir), and Nanao (the smallest reservoir), for the sake of brevity. From Figure 9, it is clear that the modeled streamflow deviate gradually from the 1:1 line and the forecast skill decreases with the increase of lead time under S3 as



expected, which is consistent with the statistical results shown in Table 4. In contrast, the scatters of the observed and modeled
streamflow implemented with forecasted climate variables fit well across the 1:1 line at different lead times under S5, observed
from Figure 10. The performance for Hongqiao Reservoir is affected explicitly by an extreme peak event that hit the reservoir
during the calibration period in Figure 9, which does not occur over the training set of data. This causes heavy underestimations
in the streamflow forecast. A more extended calibration period is required to improve the performance over such extreme peak
flow events. However, the BMA method performs well on this extreme peak flow in Hongqiao Reservoir at all lead times,
when the forecast climate forcing is applied as inputs. This is because the reservoirs in the Zhoushan Islands have relatively
small drainage areas, and thus the flow concentrates in a very short time after an extreme rain event.

*Figure 9 is here*

*Figure 10 is here*

We used the Monte Carlo simulation method to generate BMA ensemble forecasts. The number of simulations was set as 1000
in this study. To demonstrate the optimization of multi-reservoir operations based on the data-driven forecast models under
uncertainty, 90% confidence intervals associated with the deterministic predictions at BMA were further calculated. The
confidence interval provides more alternatives that are possibly useful to a tradeoff between multiple objectives, such as flood
control, hydropower generation, and improved navigation (Zhang et al., 2015). The interval performance metrics of $Cr$ and $D$
described in Section 2.3 were adopted to assess the performance of uncertain forecasts. Table 5 displays the averaged metrics
for all the 24 reservoirs under S3 and S5. Both indicators under S5 are superior to those under S3. The 90% streamflow interval
between the fifth and ninety-fifth percentiles of some representative reservoirs, e.g., Hongqiao, Goushan, and Nanao reservoirs,
are presented in Figure 11 and Figure 12. The results are consistent with those in Figure 9 and Figure 10. It is observed from
Figure 11 that the streamflow interval fails to capture the extreme peak flow for Hongqiao Reservoir under S3. The BMA
performs gradually worse with increasing lead times for the three reservoirs. However, in Figure 12, the red dots represent the
observed streamflow, most of which are covered by the 90% interval at both 1-day ahead and 7-days ahead. Therefore, the
forecast climate variables would be conducive to reduce the predictive uncertainty of real-time streamflow forecasting.

*Table 5 is here*

*Figure 11 is here*

*Figure 12 is here*

**4.3 Multi-objective reservoir operation optimization**

**4.3.1 Multi-objective reservoir operation optimization results**

For the short-term forecasting and reservoir operation purpose, a forecast horizon of 1-7 days ahead was chosen. The model
was operated following the MORDM approach under a rolling horizon scheme. Under parameterized MORDM, the decision
variables in the optimization problem were not the volumes of water to be transferred from Ningbo City and the remaining 24
reservoirs each day. Instead, the decision variables were the parameters of the RBF policies. The best operation was obtained



by conditioning the operating policies upon the following two input variables, e.g., the previous reservoir levels and current inflow. The multi-reservoir operation optimization using inflow forecasts was performed over one year (April 1st, 2007- March 31st, 2008) with 25 reservoirs. The period was selected to ensure that it did not cover the calibration datasets. The optimization was solved at each time step (a particular forecast horizon, e.g., 1-7 days) by applying NSGA-II to search the space of decision

variables and identify the islands' water allocation trajectories.

The optimized operations were both regulated based on deterministic and uncertain forecast inflow. To demonstrate the relationship between the conflicting objectives, a set of Pareto solutions over a 7-day horizon at different periods under S5 is given as an example, as shown in Figure 13. The optimization using the Pareto concept allows the operator to choose an appropriate solution depending on the prevailing circumstances and analyzing the tradeoff between the conflicting objectives.

In each of the plots, the water deficiency ratio of Daobei Plant and the sum of the remaining plants are plotted on the *x* and *y* axes, respectively. The color of the markers indicates the net operating costs with color ranging from red, representing low value, to blue, representing high value. Thus, an ideal solution would be located at the left corner (low value of the water deficiency ratio of Daobei Plant and the sum of the remaining three plants) of the plot and represented by a red (low net operating costs) marker. The black arrows have been added in the figure to guide the reader in understanding the directions of

optimization. Generally, the water deficiency ratio of Daobei Plant has an inverse relationship with that of the sum of the remaining plants (inverse relationship, i.e., the former decrease with the increase of the latter). In contrast, the water deficiency ratio of the remaining three plants has a positive relationship with the net costs (positive relationship, i.e., the former increase with the increase of the latter).

It is interesting to compare the performances associated with deterministic and uncertain forecasts. Uncertain conditions

(Figure 13(b)) show a much broader scale on the three objectives than deterministic conditions (Figure 13(a)). For instance, uncertain forecasts produce the water deficiency ratio of Daobei Plant, ranging from -40% to 80%, during 2007 August 12th - 2007 August 18st, while deterministic forecasts have a smaller range with a value from 30% to 100%. The water supply deficits under deterministic forecasts are due to the high demand happening in August, which can be mitigated when informing the reservoir operations with uncertain forecasts. In this way, we expect that if the ensemble streamflow forecasts are used in a

stochastic optimization scheme, the reservoir operation could be further enhanced because the optimization considers possible uncertainty provided by uncertain forecasts and thus takes advantage of correcting the influences of uncertainty.

*Figure 13 is here*

**4.3.2 Reservoir operation performance evaluation**

In general, forecasts are always useful for reservoir operations. The annual revenues, costs, and water supply reliability, were

chosen as metrics to compare the performance of the operating policies derived from different configurations. Reliability is a measure of how well the water demand for users is met in a water transfer system. In this case, reliability is expressed as a percentage. The system performances are averaged over a set of solutions. The annual values during the period from 2007 April 1st to 2008 March 31st at various configurations are provided in Table 6 with two decision horizons of 1 and 7 days. The





multi-reservoir operation based on observation is designed as a benchmark. It can be seen from Table 6 that the performance

indicators from the 1-day forecast horizon are better than those from 7-day using deterministic inflows (in the case of observed and forecasted inflows). Two scenarios (S3 and S5) with the 1-day forecast horizon show similar operating performance, which is consistent with the performance of the inflow forecast listed in Table 3. Recall again that S3 uses flow variables, antecedent precipitation, and evaporation as forecast inputs, while S5 uses flow variables as well as the antecedent and forecast climate forcing. In contrast to S3, the operating results of S5 with a 7-day forecast horizon are closest to that of the observation. This

is due to the improved inflow forecast performance under S5. However, it is depicted in Table 6 that the indicator of water supply reliability and net costs under S5 are inferior to those under S3. As for the stochastic forecasts, S5 outperforms S3 with lower net costs and approximate water supply reliability. In this case, the improved performance may not lead to improved decisions in deterministic forecasts.

*Table 6 is here*

The results obtained in Table 6 show that system performance derived from the observed inflows is inferior to that from other configurations. This finding cannot confirm the effectiveness of inflow forecasts. The reason for that is the forecast inflows may overestimate the actual inflows. For example, the mean value ($0.14$ m$^3$/s) of the observed inflow of Hongqiao Reservoir is lower than that of the forecasted inflow ($0.17$ m$^3$/s). In this case, the good performance presented in Table 6 is 'fake'. That is to say, although decision-makers can follow the strategies determined by the forecasted inflows, the system performance

should be assessed using the actual inflows (i.e., observed inflows). We further re-evaluated the operating strategies optimized from different configurations mentioned above using the observed inflows. The performance metrics were listed in Table 7. It is expected that the results can reveal the maximum efficiency and reliability that could be achieved based on accurate information. In general, the indicator values under deterministic forecasts in Table 7 are reduced compared with those in Table 6. The reason is that reservoir operating decisions in Table 6 are optimized based on a higher inflow series.

*Table 7 is here*

In terms of both deterministic and uncertain forecasts, net operating costs of S5 are improved significantly compared with that of S3, while water supply reliability is increased slightly. This result may suggest that improved forecasts are more skillful in making decisions when using forecast climate variables as inputs. We highlight that this result we obtained is specific for the Zhoushan Islands. Indeed, many studies show that higher forecast performance did not lead to better operation decisions

(Chiew et al., 2003; Goddard et al., 2010; Turner et al., 2017). However, some researchers draw the same conclusions as us. For instance, Anghileri et al. (2016) declared that inflow forecasts with accurate weather components would produce much smaller water supply deficits. Moreover, Anghileri et al. (2019) found that preprocessed forecasts (higher performance) were more valuable than the raw forecasts (less performance) regarding two operation performance metrics, i.e., mean annual revenues and spilled water volume.

There is also an interesting finding from that the operating performance upon deterministic forecasts deteriorates, while the performance upon uncertain forecasts can keep relatively stable. This implies that the use of uncertain forecasts in reservoir



operation can be more efficient and reliable than that of deterministic forecasts. The reason is that in a stochastic optimization scheme, the value could be further enhanced because the optimization could account for the total uncertainty provided by the ensemble forecasts. Similar results were obtained by Roulston and Smith (2003), who reported that the hydroelectric power

production derived from the ensemble forecasts was increased compared with the deterministic forecasts. Boucher et al. (2012) also found that stochastic forecasts outperformed deterministic ones with the lower turbinate flow, higher generation production, and less spillage during a flood period. Overall, in most cases, a noticeable improvement can be achieved through the use of the stochastic decision-making assistance tool.

We then assessed the performance metrics of water supply reliability over different seasons. It is noted in Figure 14 that the

deterministic forecasts are less skillful than the uncertain forecast when used in spring (JFM), summer (AMJ), autumn (JAS), and winter (OND) with the two forecast horizons. Although the operating performance using the deterministic forecast is lower due to its deterministic character, the main characteristics of the relationship between forecast quality and value remain unchanged. That is to say, the benefits of considering the forecasts are more significant when the forecast quality is higher. It indicates that the optimization is capable of exploiting efficient information to improve reservoir operations. In our multi-

objective optimization modeling, we would like to make the best use of water resources and maximize water supply. However, the operating performance in autumn shows a lower value with respect to that in other seasons. This is because the water demand in autumn is usually much higher. The shortage does not imply the non-effectiveness of our proposed forecast-based management framework but is due to the limitation of available water and pies capacity.

*Figure 14 is here*

### 4.3.3 Reservoir operation performance with different forecast horizons

The impact of different forecast horizons on the operation performance was further evaluated under different configurations, as shown in Figure 15. It is noted that the operating policy optimized from uncertain forecast inflows upon S5 outperforms that from S3. In terms of deterministic conditions, S5 improves the operation on the metrics of water supply reliability of Daobei Plant, water supply reliability of the other plants, and net costs with a variation of 2.11~13.58%, 2.74~7.38%, and -

19.94~-10.30%, respectively, compared with S3. As for uncertain conditions, S5 improves by 0.24~1.90%, 0.06~1.32%, and -59.45~-176.19%, respectively. Although the increments in water supply reliability are not insignificant, S5 can secure water demand with much less operating costs than S3, which decision makers value most. Furthermore, uncertain forecasts produce an improved ratio of 31.52~65.01%, 19.98~46.60%, -116.45~-56.95% than deterministic forecasts regarding the three metrics, respectively. Our results again highlight that uncertain forecasts are more valuable than deterministic forecasts when designing

the forecast-informed reservoir operations.

*Figure 15 is here*

With an increase in forecast horizon from 1 to 7 days, the performance in water supply reliability and net operating costs upon deterministic conditions are generally reduced. This suggests that considering a longer forecast horizon (up to 7 days) does not





necessarily improve reservoir operation without future forecast climate variables as inputs (low forecast quality). The reduced
performance in water supply reliability might be due to the fact that the optimization explores strategies to secure the whole
water demand in a longer-horizon, which results in reliability sacrifice on some particular days. This result is similar to the
finding proposed in Xu et al. (2014) who argued that the use of longer-horizon (an efficient forecast horizon longer than one
day) inflows cannot improve hydropower performance when they set the forecast horizon as one to five days. Nevertheless,
the increasing forecast horizon may not generate improved or decreased water supply reliability in uncertain conditions.
Approximate water supply volume can lead to similar revenues or fees paid to the government and managers (water fees and
water resources fees). Accordingly, the growing trend in net costs is caused by the increased operating costs, mainly dominated
by electricity prices, when the multi-reservoir is operated to supply the demand in a longer-horizon. In this case, the operation
performance varies at different conditions. This demonstrates that the relationship between forecast horizon and reservoir
operation is rather complex and depends not only on the configurations (i.e., inflow forecast quality and uncertainty) used to
determine operating rules but also on the performance metrics used to assess operation.

Our work suffers from some limitations, which could be overcome in future studies. One of the limitations is that we used the
average observed price to calculate the revenues and operating costs. In an operational and deregulated market setting, the
prices may fluctuate significantly (Anghileri et al., 2019). For instance, forecasting electricity prices is likely to improve
significantly short-term operation efficiency. The combined effects of price and streamflow forecasts on water resource
allocation are worth investigating in future studies. Another limitation is that instead of the short-term weather forecasts from
the Global Forecast System (GFS) or European Centre for Medium‐Range Weather Forecasts (ECMWF) model (Choong
and El-Shafie, 2015; Schwanenberg et al., 2015; Peng et al., 2018; Ahmad and Hossain, 2019; Liu et al., 2019), we used the
observed weather conditions as alternatives, which may result in an overestimation in forecast quality. However, forecast
uncertainty and error that generally grow up with lead time. The usefulness of the forecast information can be reduced with
the increase of the forecast horizon, and thus the operating performance. This may influence the finding we highlight above
that the relationship between forecast horizon and reservoir operation is not constant and specific. It would be interesting to
analyze the reservoir operations performance when accounting for an ensemble numerical weather prediction.

## 5. Conclusions

In this study, we proposed an AI-based management methodology to assess forecast quality and forecast-informed reservoir
operation performance together. The approach was tested on a water resources allocation system in the Zhoushan Islands,
China. Specifically, the findings obtained are summarized below.

A data-driven reservoir inflow forecasting using ML methods (LSTM, GRU, and GWO-LSSVM) was first developed with a
comprehensive calibration-validation-testing framework. The validity of the deterministic forecast was demonstrated by
applying it over 25 reservoirs with varying climate and hydrological characteristics. Results showed that the more variation
the streamflow has (a high COV value), the harder it was for the ML methods to learn the flow pattern when there existed not

enough input information. The forecast skill deteriorated with increasing lead times under such scenarios. However, short-term forecast climate forcing was efficient and scalable in forecasting the multi-reservoir inflow over the forecast horizon (1-7 days). LSTM and GRU models generated comparable performance under different configurations. Given that GRU has simpler structures and fewer parameters and required less time for modeling, it might be the preferred method for streamflow
forecasts than LSTM.

Then we used BMA to generate stochastic inflow scenarios for quantifying uncertainty based on LSTM, GRU, and GWO-LSSVM deterministic forecasts. The results demonstrated that it was difficult to conclude which individual model provided the best prediction, but the BMA did display better forecast skill in comparison to the individual ones. Including one scenario with antecedent conditions and one scenario with both antecedent and forecast information, two input combination scenarios
were compared on the uncertain forecast performance in detail. The comparison indicated that forecast climate variables would help reduce the predictive uncertainty of short-term streamflow forecasting.

The forecasting scheme was further coupled with a multi-objective reservoir operation model to optimize water resources allocation. Using a MORDM approach, we identified strategies that tradeoff between water supply reliability and operating costs in the Zhoushan Islands. A rolling horizon scheme was employed to obtain optimal operating policy over the horizon of
1-7 days. The long-term assessment over a year based on deterministic and stochastic forecasts showed quite different performances in terms of water supply reliability and net operating costs. Our averaged annual results showed that uncertain forecasts were more valuable than deterministic forecasts. The operating benefits of considering the forecasts were more significant when the forecast quality was higher. Similar results could be obtained at a seasonal scale. While showing the unquestionable benefit of implementing forecast-based reservoir operations, our results also demonstrated that the relationship
between forecast horizon and reservoir operation was complex and depended on the operating configurations (forecast quality and uncertainty) and performance measures for the Zhoushan Islands system.

Overall, the developed AI-based management framework has demonstrated a clear advantage in quantifying the uncertainties of inflow forecasts to improve the overall system performance of water allocation systems. Such a framework can be further applied to other study sites with similar problems. However, the results we obtained in this study are only specific for the
Zhoushan Islands and should be exported with care to other study sites.

*Data availability.* The data used to support the findings of this study are available from the corresponding author upon request.

*Author contribution.* Yuxue Guo, Yue-Ping Xu, and Xinting Yu designed all the experiments. Hao Chen and Haiting Gu collected and preprocessed the data. Yuxue Guo and Xinting Yu conducted all the experiments and analyzed the results. Yuxue



*Guo wrote the first draft of the manuscript with contributions from Jingkai Xie. Yue-Ping Xu supervised the study and edited*

*the manuscript.*

*Conflicts of interest. The authors declare that they have no conflict of interest.*

*Acknowledgments. This study is financially supported by the Key Project of Zhejiang Natural Science Foundation (LZ20E090001), the Design of Decision Making Support Systems for Water Diversion Project Phase III in Zhoushan City (ZSDYS-SX-SG-001) and the National Key Research and Development Plan "Inter-governmental Cooperation in International*

*Scientific and Technological Innovation" (2016YFE0122100).*

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





**List of Figure Captions**

Figure 1: Framework of the AI-based management methodology.

Figure 2: Structure of an (a) LSTM and (b) GRU cell.

Figure 3: Location of the Zhoushan Islands.

Figure 4: Schematic diagram of the Zhoushan Islands.

Figure 5: NSE values at lead times of 1 to 7 days plotted against the coefficient of variation (COV) for all the 24 reservoirs during the period of (a) calibration, (b) validation, and (c) test under S3.

Figure 6: NSE values at lead times of 1 to 7 days plotted against the coefficient of variation (COV) for all the 24 reservoirs during the period of (a) calibration, (b) validation, and (c) test S5.

Figure 7: Weights of three individual forecast models for the BMA model for all reservoirs under (a) S3 and (b) S5.

Figure 8: Improvement rates in terms of averaged (a) NSE, (b) RMSE, and (c) MAE of the BMA model for forecasts as compared with the three individual models.

Figure 9: Forecast results of (a) Hongqiao, (b) Goushan, and (c) Nanao reservoirs under S3.

Figure 10: Forecast results of (a) Hongqiao, (b) Goushan, and (c) Nanao reservoirs under S5.

Figure 11: 90% streamflow interval of the BMA method under S3.

Figure 12: 90% streamflow interval of the BMA method under S5.

Figure 13: A set of Pareto solutions at different periods over a 7-day horizon under (a) deterministic and (b) uncertain forecasts.

Figure 14: Seasonal system performance of water supply reliability.

Figure 15: Annual system performance with different forecast horizons.



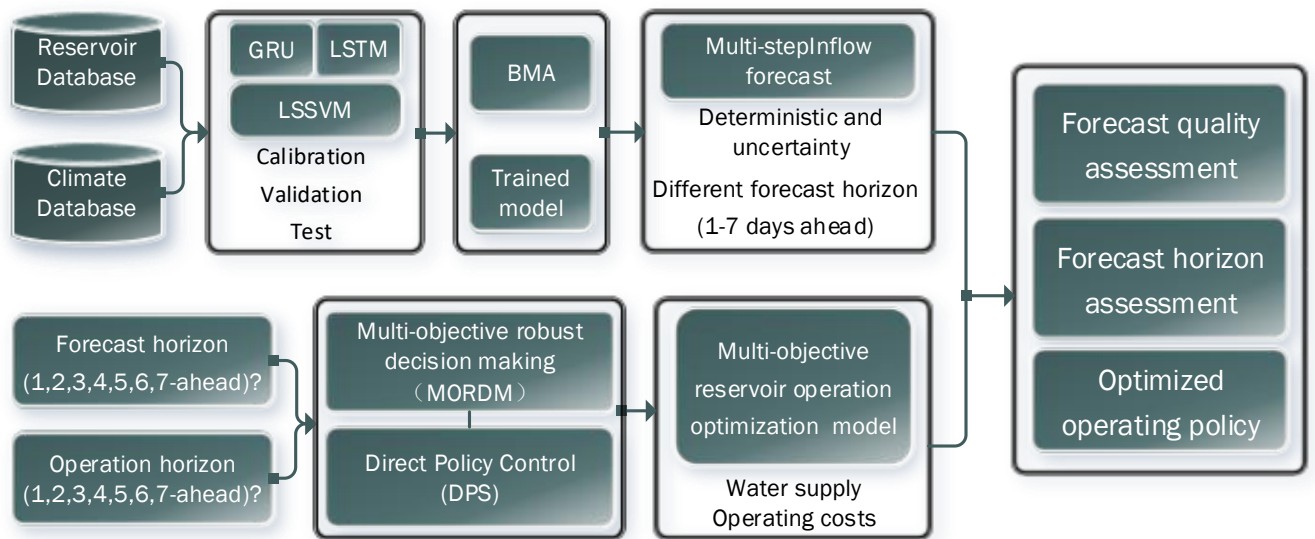


**Figure 1: Framework of the AI-based management methodology.**


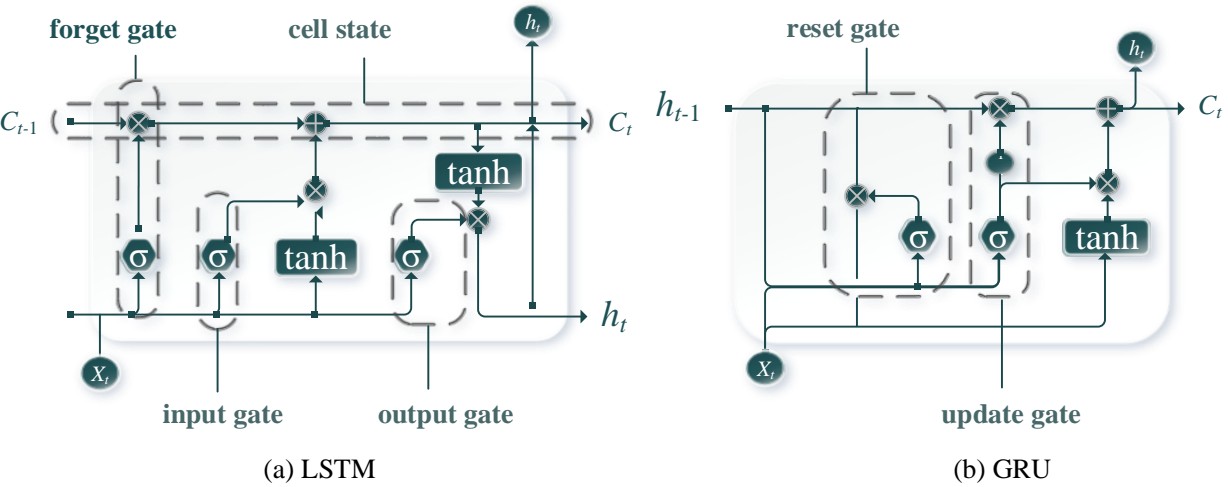

(a) LSTM

(b) GRU

**Figure 2: Structure of an (a) LSTM and (b) GRU cell.**



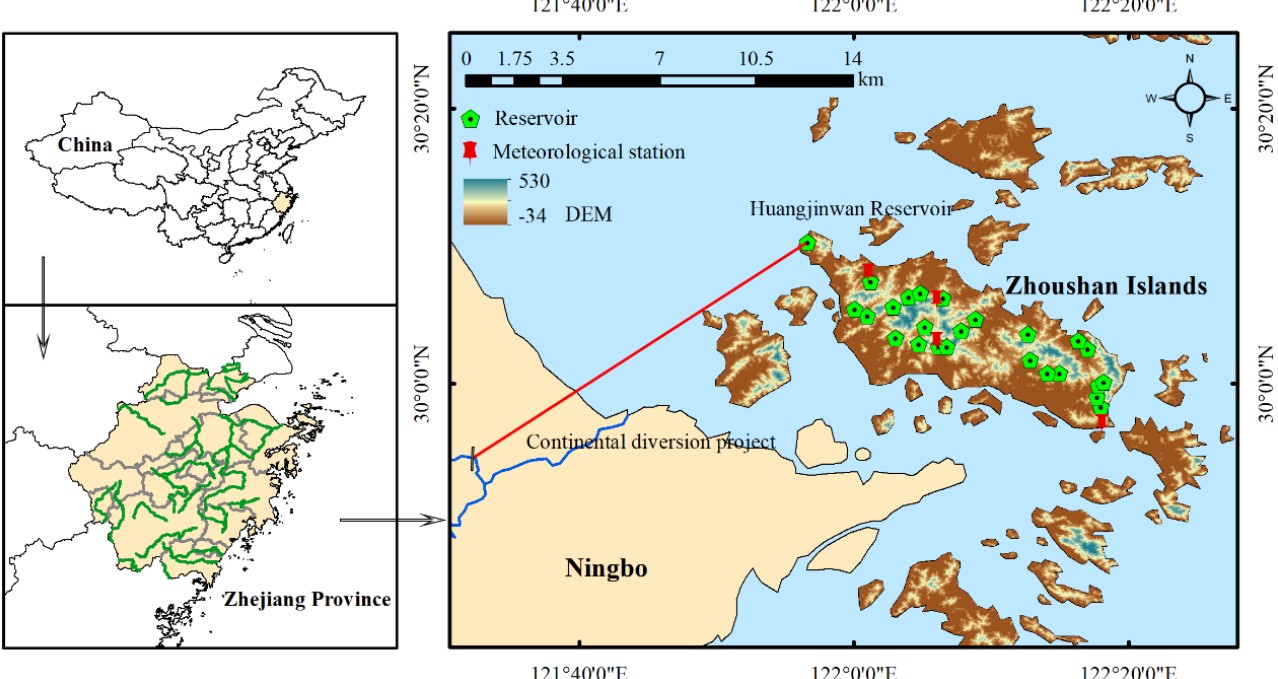

**Figure 3: Location of the Zhoushan Islands.**



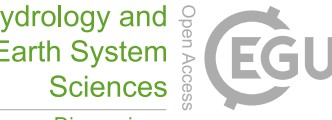


Figure 4: Schematic diagram of the Zhoushan Islands.





**Figure 5: NSE values at lead times of 1 to 7 days plotted against the coefficient of variation (COV) for all the 24 reservoirs during the period of (a) calibration, (b) validation, and (c) test under S3.**





**Figure 6: NSE values at lead times of 1 to 7 days plotted against the coefficient of variation (COV) for all the 24**
**reservoirs during the period of (a) calibration, (b) validation, and (c) test under S5.**



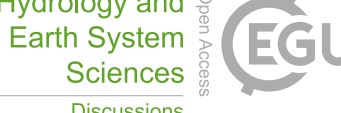

**Figure 7: Weights of three individual forecast models for the BMA model for all reservoirs under (a) S3 and (b) S5.**





**Figure 8: Improvement rates in terms of averaged (a) NSE, (b) RMSE, and (c) MAE of the BMA model for forecasts**
**as compared with the three individual models.**



**Figure 9: Forecast results of (a) Hongqiao, (b) Goushan, and (c) Nanao reservoirs under S3.**





**Figure 10: Forecast results of (a) Hongqiao, (b) Goushan, and (c) Nanao reservoirs under S5.**




**Figure 11: 90% streamflow interval of the BMA method under S3.**






Figure 12: 90% streamflow interval of the BMA method under S5.



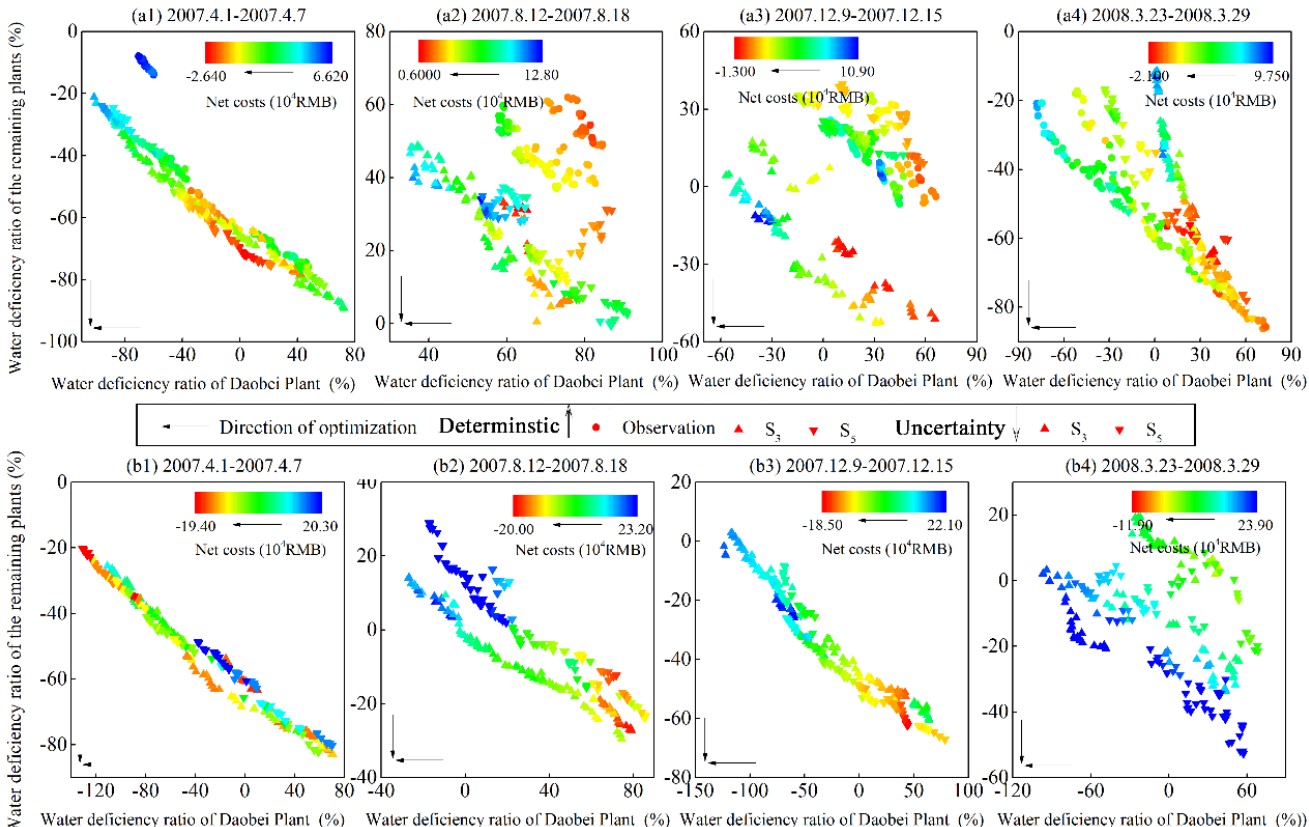

**Figure 13: A set of Pareto solutions at different periods over a 7-day horizon under (a) deterministic and (b) uncertain forecasts.**





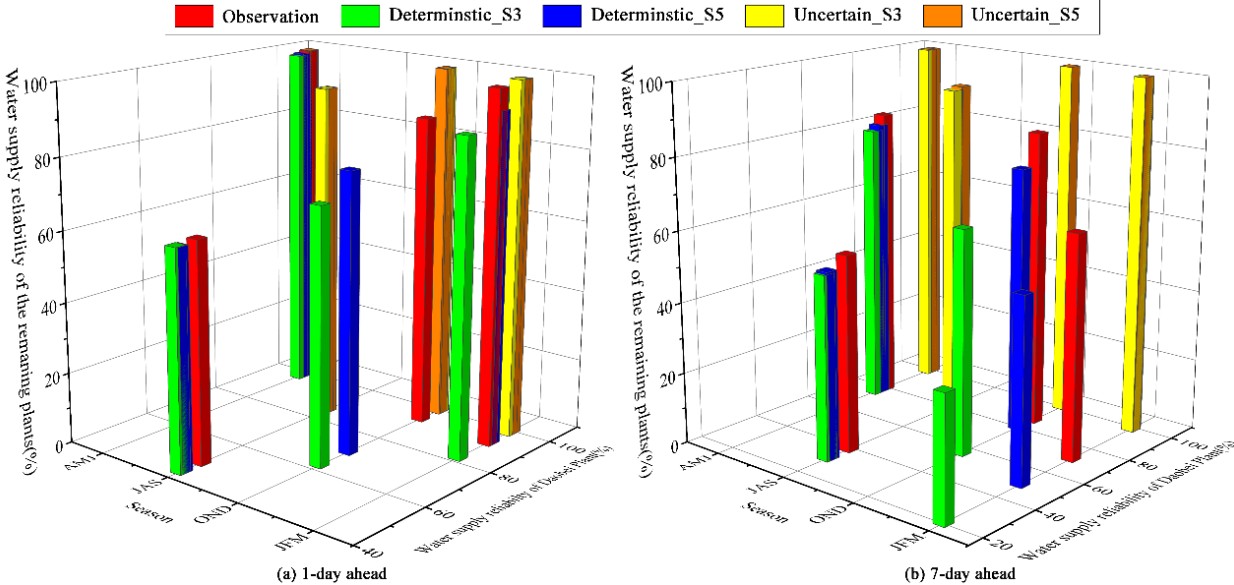

**Figure 14: Seasonal system performance of water supply reliability.**






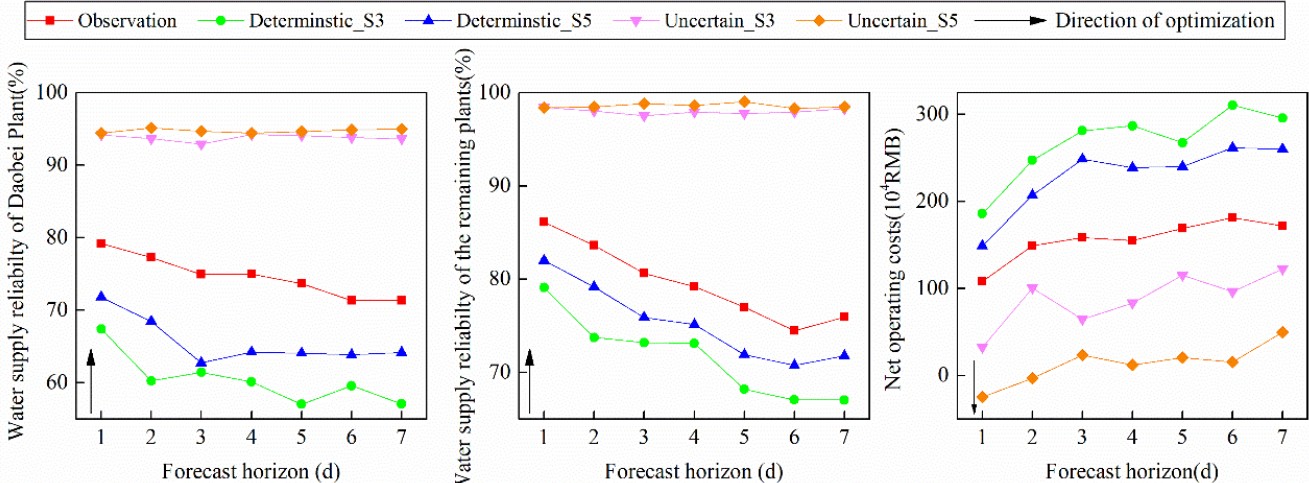

**Figure 15: Annual system performance with different forecast horizons.**





**List of Table Captions**

Table 1: Reservoir characteristics in the Zhoushan Islands.

Table 2: Five input combination scenarios.

Table 3: NSE ranges ([min, max]) for all reservoirs with the different configurations during the calibration, validation, and test periods.

Table 4: Performance metric ranges ([min, max]) for all 24 reservoirs of BMA methods under S3 and S5.

Table 5: Ranges of interval performance metrics ([min, max]) for all the 24 reservoirs under S3 and S5.

Table 6: Annual system performance using forecast inflow information.

Table 7: Annual system performance using observed inflow information.



**Table 1: Reservoir characteristics in the Zhoushan Islands.**

| District | Reservoir name | Reservoir storage ($10^4$m$^3$) | Dead storage ($10^4$m$^3$) | Normal storage ($10^4$m$^3$) | Drainage area (km$^2$) | Mean flow (m$^3$/s) | Standard deviation of flow (m$^3$/s) | COV |
|---|---|---|---|---|---|---|---|---|
| Hongqiao | Hongqiao | 1307 | 12 | 1015 | 13.4 | 0.15 | 0.77 | 5.08 |
| | Chahe | 254 | 35.08 | 185 | 8 | 0.11 | 0.49 | 4.34 |
| | Hongwei | 85 | 36 | 76.1 | 1.94 | 0.10 | 0.21 | 2.06 |
| | Chengbei | 123 | 45 | 111.1 | 4.98 | 0.10 | 0.33 | 3.34 |
| | Mahuangshan | 354 | 17.15 | 286.4 | 4.87 | 0.10 | 0.32 | 3.28 |
| | Xiamen | 281 | 42 | 240 | 4 | 0.10 | 0.29 | 2.92 |
| | Cenggang | 733 | 14.2 | 627 | 6.6 | 0.10 | 0.41 | 3.91 |
| | Longtan | 160 | 9 | 133.6 | 2.27 | 0.10 | 0.22 | 2.18 |
| Daobei | Dongaonong | 185 | 3.4 | 159.84 | 2.6 | 0.10 | 0.23 | 2.28 |
| | Changmenli | 205 | 49.49 | 179.5 | 2.3 | 0.10 | 0.22 | 2.17 |
| | Tuanjie | 122 | 30.4 | 106.6 | 2.05 | 0.10 | 0.21 | 2.09 |
| | Changchunling | 410 | 34.3 | 368.3 | 5.41 | 0.10 | 0.35 | 3.53 |
| | Yaojiawan | 124 | 31.09 | 105 | 1.46 | 0.10 | 0.22 | 2.23 |
| | Jinlin | 154 | 40.48 | 125.9 | 2.42 | 0.10 | 0.20 | 1.96 |
| | BaiquanLing | 204 | 12.56 | 177.4 | 3 | 0.10 | 0.24 | 2.44 |
| | Chenao | 236 | 59 | 195.2 | 4.13 | 0.10 | 0.29 | 2.99 |
| Dongbu | Dashiao | 293 | 49.1 | 254 | 2.8 | 0.10 | 0.23 | 2.37 |
| | Pingdi | 317 | 1 | 317.2 | 0.6 | 0.10 | 0.19 | 1.87 |
| | Dongao | 457.8 | 47.5 | 384 | 6.4 | 0.10 | 0.40 | 3.87 |
| | Goushan | 194 | 4.59 | 170 | 2.73 | 0.10 | 0.23 | 2.34 |
| | Nanao | 73 | 15.8 | 66.7 | 1.21 | 0.10 | 0.20 | 1.94 |
| | Ludong | 142 | 54 | 118.5 | 3.7 | 0.10 | 0.27 | 2.75 |
| | Yingjiawan | 124 | 31.09 | 105 | 1.46 | 0.10 | 0.31 | 3.17 |
| | Shatianao | 127 | 20.8 | 116.5 | 4.54 | 0.10 | 0.19 | 1.89 |




**Table 2: Five input combination scenarios.**

| ID | Scenario | Input combination |
|---|---|---|
| 1 | S1 | $Q_a$ |
| 2 | S2 | $P_a, E_a$ |
| 3 | S3 | $Q_a, P_a, E_a$ |
| 4 | S4 | $P_a, P_f, E_a, E_f$ |
| 5 | S5 | $Q_a, P_a, P_f, E_a, E_f$ |





**Table 3: NSE ranges ([min, max]) for all reservoirs with the different configurations during the calibration, validation, and test periods.**

| Period | Forecast horizon (d) | LSTM | | | | | GRU | | | | | GWO-LSSVM | | | | |
|---|---|---|---|---|---|---|---|---|---|---|---|---|---|---|---|---|
| | | S1 | S2 | S3 | S4 | S5 | S1 | S2 | S3 | S4 | S5 | S1 | S2 | S3 | S4 | S5 |
| Calibration | 1 | [0.11, 0.87] | [0.57, 0.94] | [0.61, 0.97] | [0.8, 0.96] | [0.93, 0.99] | [0.18, 0.87] | [0.57, 0.87] | [0.53, 0.98] | [0.66, 0.96] | [0.89, 0.99] | [0.17, 0.91] | [0.54, 0.86] | [0.58, 0.97] | [0.87, 0.97] | [0.96, 0.99] |
| | 2 | [0.05, 0.58] | [0.27, 0.72] | [0.34, 0.93] | [0.82, 0.95] | [0.91, 0.99] | [0.07, 0.58] | [0.29, 0.66] | [0.36, 0.87] | [0.78, 0.96] | [0.87, 0.99] | [0.08, 0.58] | [0.27, 0.72] | [0.31, 0.83] | [0.87, 0.97] | [0.94, 0.97] |
| | 3 | [0.03, 0.48] | [0.11, 0.55] | [0.13, 0.63] | [0.75, 0.94] | [0.91, 0.98] | [0.05, 0.51] | [0.10, 0.52] | [0.14, 0.62] | [0.79, 0.95] | [0.92, 0.98] | [0.05, 0.51] | [0.13, 0.55] | [0.11, 0.59] | [0.86, 0.94] | [0.93, 0.96] |
| | 4 | [0.03, 0.44] | [0.08, 0.49] | [0.10, 0.56] | [0.84, 0.95] | [0.94, 0.98] | [0.04, 0.45] | [0.08, 0.45] | [0.12, 0.56] | [0.80, 0.95] | [0.90, 0.98] | [0.05, 0.45] | [0.1, 0.8] | [0.09, 0.54] | [0.87, 0.92] | [0.92, 0.95] |
| | 5 | [0.01, 0.17] | [0.02, 0.16] | [0.03, 0.22] | [0.74, 0.95] | [0.94, 0.98] | [0.02, 0.17] | [0.02, 0.17] | [0.05, 0.22] | [0.86, 0.95] | [0.89, 0.98] | [0.03, 0.16] | [0.05, 0.46] | [0.03, 0.23] | [0.87, 0.93] | [0.93, 0.95] |
| | 6 | [0.01, 0.39] | [0.06, 0.39] | [0.07, 0.44] | [0.83, 0.95] | [0.93, 0.98] | [0.02, 0.4] | [0.05, 0.38] | [0.09, 0.46] | [0.8, 0.95] | [0.91, 0.98] | [0.03, 0.41] | [0.07, 0.87] | [0.05, 0.45] | [0.87, 0.90] | [0.89, 0.94] |
| | 7 | [0.01, 0.18] | [0.04, 0.19] | [0.04, 0.24] | [0.84, 0.96] | [0.94, 0.97] | [0.02, 0.19] | [0.04, 0.19] | [0.07, 0.26] | [0.86, 0.95] | [0.93, 0.97] | [0.02, 0.19] | [0.06, 0.81] | [0.06, 0.25] | [0.84, 0.88] | [0.85, 0.94] |
| Validation | 1 | [0.09, 0.90] | [0.45, 0.93] | [0.50, 0.92] | [0.79, 0.96] | [0.82, 0.97] | [0.11, 0.87] | [0.47, 0.87] | [0.51, 0.98] | [0.34, 0.96] | [0.81, 0.99] | [0.04, 0.79] | [0.5, 0.95] | [0.58, 0.88] | [0.70, 0.93] | [0.76, 0.90] |
| | 2 | [0.08, 0.85] | [0.01, 0.87] | [0.01, 0.90] | [0.42, 0.95] | [0.64, 0.95] | [0.09, 0.58] | [0.09, 0.66] | [0.07, 0.87] | [0.54, 0.96] | [0.76, 0.99] | [0.00, 0.74] | [0.01, 0.83] | [0.03, 0.86] | [0.70, 0.93] | [0.67, 0.95] |
| | 3 | [0.08, 0.83] | [0.02, 0.83] | [-0.01, 0.87] | [0.79, 0.96] | [0.68, 0.96] | [0.09, 0.51] | [0.09, 0.52] | [0.08, 0.62] | [0.52, 0.95] | [0.77, 0.98] | [0.00, 0.74] | [0.02, 0.8] | [0.03, 0.83] | [0.74, 0.94] | [0.76, 0.95] |
| | 4 | [0.08, 0.83] | [0.01, 0.84] | [0.02, 0.89] | [0.80, 0.96] | [0.68, 0.95] | [0.09, 0.45] | [0.08, 0.45] | [0.07, 0.56] | [0.52, 0.95] | [0.78, 0.98] | [0.00, 0.74] | [0.01, 0.81] | [0.04, 0.84] | [0.73, 0.94] | [0.77, 0.95] |
| | 5 | [0.08, 0.81] | [0.01, 0.82] | [0.01, 0.85] | [0.75, 0.96] | [0.67, 0.96] | [0.08, 0.17] | [0.07, 0.17] | [0.06, 0.22] | [0.52, 0.95] | [0.74, 0.98] | [0.00, 0.72] | [-0.01, 0.78] | [0.02, 0.81] | [0.70, 0.94] | [0.76, 0.94] |
| | 6 | [0.08, 0.80] | [0.00, 0.80] | [0.00, 0.84] | [0.80, 0.95] | [0.67, 0.94] | [0.09, 0.4] | [0.07, 0.38] | [0.05, 0.46] | [0.51, 0.95] | [0.80, 0.98] | [0.01, 0.71] | [0.00, 0.77] | [0.02, 0.79] | [0.73, 0.94] | [0.76, 0.94] |
| | 7 | [0.07, 0.78] | [0.01, 0.79] | [0.00, 0.82] | [0.76, 0.96] | [0.69, 0.95] | [0.08, 0.19] | [0.07, 0.19] | [0.06, 0.26] | [0.53, 0.95] | [0.76, 0.97] | [0.00, 0.70] | [0.00, 0.76] | [0.02, 0.79] | [0.77, 0.95] | [0.77, 0.95] |
| Test | 1 | [-0.04, 0.69] | [0.50, 0.73] | [0.56, 0.89] | [0.58, 0.77] | [0.54, 0.87] | [-0.09, 0.71] | [0.48, 0.74] | [0.54, 0.87] | [0.54, 0.76] | [0.65, 0.89] | [0.04, 0.72] | [0.53, 0.71] | [0.58, 0.88] | [0.69, 0.79] | [0.76, 0.90] |
| | 2 | [-0.13, 0.69] | [0.04, 0.62] | [0.03, 0.75] | [0.41, 0.78] | [0.63, 0.85] | [-0.16, 0.66] | [0.04, 0.59] | [0.03, 0.7] | [0.58, 0.77] | [0.61, 0.86] | [-0.01, 0.71] | [0.03, 0.63] | [0.03, 0.78] | [0.69, 0.79] | [0.75, 0.86] |
| | 3 | [0.01, 0.20] | [0.48, 0.65] | [0.50, 0.72] | [0.58, 0.73] | [0.56, 0.83] | [0.03, 0.2] | [0.51, 0.65] | [0.48, 0.71] | [0.61, 0.78] | [0.69, 0.82] | [0.03, 0.2] | [0.50, 0.63] | [0.58, 0.70] | [0.67, 0.74] | [0.73, 0.81] |
| | 4 | [-0.01, 0.15] | [0.02, 0.13] | [0.01, 0.17] | [0.41, 0.77] | [0.63, 0.83] | [-0.02, 0.13] | [0.02, 0.13] | [0.01, 0.15] | [0.57, 0.77] | [0.60, 0.83] | [-0.01, 0.13] | [0.01, 0.12] | [0.02, 0.18] | [0.68, 0.81] | [0.74, 0.83] |
| | 5 | [-0.01, 0.14] | [0.01, 0.14] | [-0.01, 0.17] | [0.63, 0.78] | [0.63, 0.81] | [-0.02, 0.14] | [-0.01, 0.15] | [0.00, 0.18] | [0.53, 0.75] | [0.62, 0.80] | [0.00, 0.12] | [0.01, 0.15] | [0.01, 0.21] | [0.65, 0.77] | [0.69, 0.80] |
| | 6 | [-0.01, 0.05] | [0.01, 0.08] | [0.00, 0.10] | [0.62, 0.76] | [0.62, 0.80] | [-0.01, 0.05] | [0.01, 0.08] | [0.00, 0.08] | [0.56, 0.82] | [0.58, 0.82] | [-0.01, 0.04] | [0.02, 0.07] | [0.01, 0.08] | [0.59, 0.75] | [0.62, 0.77] |





| 7 | [-0.02, 0.04] | [0.23, 0.40] | [0.19, 0.45] | [0.57, 0.74] | [0.55, 0.82] | [-0.02, 0.03] | [0.22, 0.47] | [0.20, 0.47] | [0.59, 0.81] | [0.65, 0.79] | [-0.01, 0.03] | [0.24, 0.34] | [0.27, 0.40] | [0.69, 0.82] | [0.75, 0.81] |




**Table 4: Performance metric ranges ([min, max]) for all 24 reservoirs of BMA methods under S3 and S5.**

| Period | Forecast horizon (d) | NSE | | RMSE(m³/s) | | MAE(m³/s) | |
|---|---|---|---|---|---|---|---|
| | | S3 | S5 | S3 | S5 | S3 | S5 |
| Calibration | 1 | [0.60, 0.98] | [0.97, 0.99] | [0.02, 0.36] | [0.01, 0.09] | [0.01, 0.09] | [0.01, 0.03] |
| | 2 | [0.36, 0.92] | [0.96, 0.99] | [0.04, 0.46] | [0.02, 0.10] | [0.02, 0.13] | [0.01, 0.04] |
| | 3 | [0.14, 0.63] | [0.95, 0.98] | [0.09, 0.53] | [0.02, 0.11] | [0.03, 0.15] | [0.01, 0.04] |
| | 4 | [0.12, 0.57] | [0.94, 0.98] | [0.10, 0.54] | [0.02, 0.12] | [0.04, 0.17] | [0.01, 0.04] |
| | 5 | [0.04, 0.23] | [0.95, 0.98] | [0.14, 0.56] | [0.02, 0.12] | [0.04, 0.17] | [0.01, 0.04] |
| | 6 | [0.07, 0.46] | [0.94, 0.98] | [0.11, 0.55] | [0.02, 0.10] | [0.04, 0.17] | [0.01, 0.04] |
| | 7 | [0.06, 0.25] | [0.94, 0.97] | [0.13, 0.52] | [0.03, 0.11] | [0.05, 0.15] | [0.01, 0.04] |
| Validation | 1 | [0.60, 0.92] | [0.84, 0.96] | [0.08, 0.66] | [0.06, 0.39] | [0.02, 0.13] | [0.02, 0.10] |
| | 2 | [0.07, 0.93] | [0.80, 0.96] | [0.07, 1.09] | [0.06, 0.33] | [0.03, 0.19] | [0.02, 0.09] |
| | 3 | [0.06, 0.90] | [0.82, 0.95] | [0.09, 1.09] | [0.06, 0.30] | [0.04, 0.21] | [0.02, 0.09] |
| | 4 | [0.08, 0.91] | [0.85, 0.96] | [0.08, 1.07] | [0.06, 0.34] | [0.04, 0.19] | [0.02, 0.09] |
| | 5 | [0.09, 0.85] | [0.85, 0.96] | [0.11, 1.08] | [0.05, 0.29] | [0.05, 0.22] | [0.02, 0.09] |
| | 6 | [0.06, 0.83] | [0.86, 0.95] | [0.11, 1.08] | [0.06, 0.34] | [0.05, 0.21] | [0.03, 0.09] |
| | 7 | [0.04, 0.82] | [0.87, 0.96] | [0.12, 1.10] | [0.06, 0.35] | [0.05, 0.22] | [0.02, 0.10] |
| Test | 1 | [0.60, 0.89] | [0.76, 0.89] | [0.08, 0.68] | [0.08, 0.47] | [0.03, 0.20] | [0.03, 0.16] |
| | 2 | [0.05, 0.76] | [0.68, 0.87] | [0.12, 1.05] | [0.09, 0.50] | [0.05, 0.27] | [0.04, 0.16] |
| | 3 | [0.59, 0.73] | [0.68, 0.83] | [0.13, 0.69] | [0.10, 0.53] | [0.05, 0.23] | [0.04, 0.16] |
| | 4 | [0.03, 0.18] | [0.69, 0.83] | [0.22, 1.06] | [0.10, 0.54] | [0.08, 0.29] | [0.04, 0.16] |
| | 5 | [0.01, 0.21] | [0.68, 0.81] | [0.21, 1.08] | [0.11, 0.51] | [0.08, 0.30] | [0.04, 0.16] |
| | 6 | [0.02, 0.09] | [0.64, 0.81] | [0.23, 1.07] | [0.11, 0.63] | [0.09, 0.30] | [0.04, 0.18] |
| | 7 | [0.25, 0.43] | [0.67, 0.80] | [0.19, 0.84] | [0.12, 0.55] | [0.09, 0.28] | [0.05, 0.16] |





**Table 5: Ranges of interval performance metrics ([min, max]) for all the 24 reservoirs under S3 and S5.**

| Period | Forecast horizon (d) | Cr(%) S3 | Cr(%) S5 | $D$(m³/s) S3 | $D$(m³/s) S5 |
|---|---|---|---|---|---|
| Calibration | 1 | [94.36, 99.86] | [96.29, 99.86] | [0.01, 0.06] | [0.01, 0.03] |
| | 2 | [93.67, 99.17] | [94.91, 99.66] | [0.01, 0.05] | [0.01, 0.03] |
| | 3 | [94.36, 98.21] | [95.05, 99.59] | [0.01, 0.04] | [0.01, 0.03] |
| | 4 | [92.98, 96.97] | [95.67, 99.72] | [0.02, 0.04] | [0.01, 0.04] |
| | 5 | [93.26, 96.22] | [94.57, 99.79] | [0.01, 0.04] | [0.01, 0.04] |
| | 6 | [93.26, 96.70] | [95.74, 99.66] | [0.02, 0.05] | [0.01, 0.04] |
| | 7 | [92.64, 96.15] | [95.53, 99.72] | [0.02, 0.05] | [0.01, 0.03] |
| Validation | 1 | [92.88, 99.73] | [96.44, 100.00] | [0.01, 0.05] | [0.01, 0.03] |
| | 2 | [94.25, 99.45] | [93.97, 100.00] | [0.01, 0.05] | [0.01, 0.03] |
| | 3 | [92.33, 97.26] | [94.52, 100.00] | [0.01, 0.02] | [0.01, 0.04] |
| | 4 | [92.60, 96.16] | [93.42, 99.73] | [0.01, 0.06] | [0.01, 0.04] |
| | 5 | [91.78, 94.79] | [95.07, 100.00] | [0.01, 0.04] | [0.01, 0.04] |
| | 6 | [91.78, 94.79] | [95.07, 100.00] | [0.01, 0.04] | [0.01, 0.04] |
| | 7 | [90.68, 93.42] | [93.70, 99.73] | [0.00, 0.03] | [0.01, 0.03] |
| Test | 1 | [90.83, 99.32] | [93.84, 99.73] | [0.03, 0.22] | [0.03, 0.15] |
| | 2 | [92.75, 97.95] | [94.25, 99.73] | [0.04, 0.26] | [0.03, 0.16] |
| | 3 | [92.48, 97.40] | [94.39, 99.73] | [0.05, 0.26] | [0.03, 0.15] |
| | 4 | [91.66, 95.35] | [94.39, 99.59] | [0.07, 0.28] | [0.04, 0.16] |
| | 5 | [90.70, 94.12] | [94.66, 99.45] | [0.07, 0.29] | [0.04, 0.15] |
| | 6 | [90.83, 93.98] | [93.43, 99.45] | [0.08, 0.29] | [0.05, 0.18] |
| | 7 | [89.88, 92.48] | [93.57, 99.45] | [0.08, 0.28] | [0.04, 0.16] |





**Table 6:** **Annual system performance using forecast inflow information.**

| Inflow Configuration | | Forecast horizon (d) | Revenues ($10^4$RMB) | Costs ($10^4$RMB) | Net costs ($10^4$RMB) | Reliability (%) Daobei | Others |
|---|---|---|---|---|---|---|---|
| Observation | | 1 | 3228.38 | 3336.52 | 108.15 | 79.22 | 86.10 |
| | | 7 | 2651.01 | 2822.95 | 171.94 | 71.31 | 75.94 |
| Deterministic | S3 | 1 | 3541.27 | 3633.50 | 92.23 | 79.68 | 96.90 |
| | S5 | 1 | 3596.23 | 3690.32 | 94.09 | 79.59 | 96.31 |
| | S3 | 7 | 3262.51 | 3401.53 | 139.02 | 80.93 | 91.84 |
| | S5 | 7 | 2945.42 | 3118.95 | 173.53 | 76.10 | 85.27 |
| Uncertain | S3 | 1 | 3931.58 | 3837.88 | -93.70 | 93.49 | 99.80 |
| | S5 | 1 | 3988.70 | 3791.54 | -197.16 | 92.63 | 99.58 |
| | S3 | 7 | 3946.61 | 3902.59 | -44.02 | 94.03 | 100.00 |
| | S5 | 7 | 3911.55 | 3846.44 | -65.11 | 92.83 | 99.38 |



**Table 7: Annual system performance using observed inflow information.**

| Inflow Configurations | | Forecast horizon (d) | Revenues ($10^4$RMB) | Costs ($10^4$RMB) | Net costs ($10^4$RMB) | Reliability (%) | |
|---|---|---|---|---|---|---|---|
| | | | | | | Daobei | Others |
| Observation | | 1 | 3228.38 | 3336.52 | 108.15 | 79.22 | 86.10 |
| | | 7 | 2651.01 | 2822.95 | 171.94 | 71.31 | 75.94 |
| Deterministic | S3 | 1 | 2597.17 | 2783.31 | 186.14 | 67.42 | 79.10 |
| | S5 | 1 | 2735.64 | 2884.66 | 149.02 | 71.79 | 82.00 |
| | S3 | 7 | 2159.05 | 2454.79 | 295.75 | 57.08 | 67.04 |
| | S5 | 7 | 2371.45 | 2631.57 | 260.11 | 64.15 | 71.80 |
| Uncertain | S3 | 1 | 3788.08 | 3820.27 | 32.18 | 94.18 | 98.45 |
| | S5 | 1 | 3805.98 | 3781.46 | -24.52 | 94.42 | 98.38 |
| | S3 | 7 | 3762.07 | 3884.75 | 122.68 | 93.64 | 98.28 |
| | S5 | 7 | 3785.55 | 3835.29 | 49.75 | 94.99 | 98.46 |