# Peer review of "AI-based techniques for multi-step streamflow forecasts: Application for multi-objective reservoir operation optimization and performance assessment"

_Hydrology and Earth System Sciences, 2020_

## Referee Comment (RC1) · Anonymous Referee #1 · 12 Jan 2021

**HESS-2020-617**

This work investigates a multi-objective reservoir operation problem with streamflow forecasting by combining some models and procedures well-known in water resources engineering. The theory and method are not new, but the case studies are detailed and impressive, with some meaningful conclusions drawn in the work. The manuscript could be organized better, and the mathematical formulation could be expressed in a more professional way, so that the contributions of the work can be better appreciated by readers.

**Specific Comments:**

There are some comments, questions and suggestions that may help improve the quality of the manuscript, including:

(1) Some comments in literature review could be more precisely. The LSTM and GRU, for example, were not only applied in few previous works (refer to Line 55 in the manuscript), and the research works on impacts of forecast horizon on reservoir operation were not rare (Lines 59 and 71).

(2) It is unclear how the weight matrices involved in the forecasting models (Lines: 124 and 136) were estimated, and what / which criteria were used in calibration.

(3) It is left unexplained:
   - How the parameters used to define the operational policy are estimated,
   - What specific hydrological variables are included in the "policy inputs",
   - How these "policy inputs" are related to the decision horizon, and
   - How the policy could be implemented with all constraints enforced in a day-by-day practice,
   - Why it is called "multi-objective" since involving only an objective (26)?

(4) I think this work has formulated an incomplete reservoir operation problem. The water balance, for instance, does not reflect the hydraulic connections shown in Figure 4. The relationships between water supply, pumping flow, inflow and discharge are not incorporated in the model. Also, how the MORDM is related to this operational problem? The model looks like a linear programming problem that can be easily solved.

(5) The manuscript will benefit from more logically organizing its contents. The "Results and Discussion" are usually a part of the case studies. Theory, models, procedures and definitions are generally presented before case studies, and some of them need more detailed introduction, including:
   - How the weights in the BMA are determined (Line 320),
   - How the Monte Carlo simulation method is used to generate BMA ensemble forecasts (Line 359),
   - What "the previous water levels" is supposed to mean (Line 381),

- Why the NSGA-II are still needed since we already have the operation policy determined (Line 383),
- How the deterministic, uncertain and observed streamflow are used in the operation (399),
- How the Pareto solutions are identified (Line 387),
- Whether or not the annual revenues, costs, and water supply reliability (Line 409) are used as multiple objectives when determining the operating policy,
- "Fake" results do not have any meaningful value, so why they are included in Table 6 in the first place (Line 428).

(6) To the best of my understanding, the NSE was used to calibrate the forecasting models while the RMSE and MAE are also used in assessing the performance of the models. I think it should be a fairer practice by using multi-criteria to do both the calibration and assessment.

(7) Please justify why the Radial Basis Functions are used to parameterize the policy (Line 199)?

(8) Including the test period when minimizing the NSE (Line 285) will make it lose efficacy in assessing the model performance in future.

**Technical Corrections:**

(1) Please rewrite the term ($\sum_{i=1}^{K} w_k f_k$) in equation (19), which just does not make sense to me, with the $f_k$ being a model.

(2) Please double check all the mathematical expressions. To mention a few,
- In equations (19) and (20), the sum should be operated over subscript "$k$" rather than "$i$".
- It might not be right that the subscript "$k$" on the left side does not appear on the right side of the equation (28).
- It sounds not right to me in equation (34), where a variable without subscript "j" is summed over "j".
- It is questionable that the equation (35) does not have a subscript for the first sum operator to operate over.
- Expressing a variable subscript "$n$" in "$Q_{max\_n}$" (Line 247) is something strange.
- Please check on all similar unprofessional expressions in (39), (42) and (43).

(3) Please do not omit subscripts in mathematical symbols. And for all the definitions of math symbols, all the subscripts in any symbol should appear in its definition.

---

## Referee Comment (RC2) · Anonymous Referee #2 · 5 Mar 2021

The authors proposed a stochastic analysis based on AI for water resources optimization and localization. The paper is fully in the aims of the Journal and it is well written, well-structured and characterized by adequate language. I have a simple observation: - In Figure 5, deterministic S5 performance indicators overlap each other, I would suggest to modify color or width to improve the readability.

---

## Author Comment (AC1) · 25 Mar 2021

Dear Referee #1,

We highly appreciate your review and useful suggestions for our manuscript. We provide our responses to your queries below.

Kind regards, all authors

Queries by anonymous referee #1 RC1 & answers by authors are as follows:

**Comment #1:** Some comments in literature review could be more precisely.

- The LSTM and GRU, for example, were not only applied in few previous works (refer to Line 55 in the manuscript)

**Authors' response:** Thank you. We will modify Line 55 as "*LSTM and GRU networks have been successfully applied in many fields (Greff et al., 2017; Zhang et al., 2018; Jung et al., 2020; Shahid et al., 2020; Ayzel and Heistermann, 2021), and they are demonstrated to generate comparable performances. But GRU has a more straightforward structure and a higher operation speed than LSTM. Recently, many applications that assessed them together are also found in the hydrological field (Gao et al., 2020; Muhammad et al., 2020).*"

**References**

Ayzel, G., Heistermann, M. The effect of calibration data length on the performance of a conceptual hydrological model versus LSTM and GRU: A case study for six basins from the CAMELS dataset. Computers & Geosciences, 149, 104708, https://doi.org/10.1016/j.cageo.2021.104708, 2021.

Gao, S., Huang, Y., Zhang, S., et al. Short-term runoff prediction with GRU and LSTM networks without requiring time step optimization during sample generation. Journal of Hydrology, 589, 125188, https://doi.org/10.1016/j.jhydrol.2020.125188, 2020.

Greff, K., Srivastava, R. K., Koutník, J., et al. LSTM: A Search Space Odyssey. IEEE Transactions on Neural Networks and Learning Systems, 28(10), 2222-2232, https://doi.org/10.1109/TNNLS.2016.2582924, 2017.

Jung, Y., Jung, J., Kim, B., et al. Long short-term memory recurrent neural network for modeling temporal patterns in long-term power forecasting for solar PV facilities: Case study of South

Korea. Journal of Cleaner Production, 250, 119476, https://doi.org/10.1016/j.jclepro.2019.119476, 2020.

Muhammad A.U., Li X., Feng J. Using LSTM GRU and Hybrid Models for Streamflow Forecasting. Machine Learning and Intelligent Communications 2019. Lecture Notes of the Institute for Computer Sciences, Social Informatics and Telecommunications Engineering, Springer, 294, 510-524, https://doi.org/10.1007/978-3-030-32388-2_44, 2019.

Shahid, F., Zameer, A., Muneeb, M. Predictions for COVID-19 with deep learning models of LSTM, GRU and Bi-LSTM. Chaos, Solitons & Fractals, 140, 110212, https://doi.org/10.1016/j.chaos.2020.110212, 2020.

Zhang, D., Lindholm, G., Ratnaweera, H. Use long short-term memory to enhance Internet of Things for combined sewer overflow monitoring. Journal of Hydrology, 556, 409-418, https://doi.org/10.1016/j.jhydrol.2017.11.018, 2018.

- The research works on impacts of forecast horizon on reservoir operation were not rare (Lines 59 and 71).

**Authors' response:** Thanks for your comments.

We will modify Line 59 as "*While a considerable research effort has been made to evaluate and improve the quality of streamflow forecasts (Gibbs et al., 2018; Nanda et al., 2019; Sharma et al., 2019; Van Osnabrugge et al., 2019; Feng et al., 2020; Pechlivanidis et al., 2020), how forecasts impact decision-making in the real-time reservoir operations has gradually gained researchers' attention (Goddard et al., 2010; Shamir, 2017; Anghileri et al., 2019; Alexander et al., 2020; Hadi et al., 2020), e.g., do high-quality forecasts mean improved decision?*"

We will modify Line 71 as "*There is often a mismatch between the information needed for reservoir operations and the skillful lead time of the reservoir inflow forecast (Anghileri et al., 2016). It is crucial to demonstrate the applicability and effectiveness of the forecast horizon in a forecast-based reservoir operation system (Xu et al., 2014).*"

**References**

Alexander, S., Yang, G., Addisu, G., et al. Forecast-informed reservoir operations to guide hydropower and agriculture allocations in the Blue Nile basin, Ethiopia. International Journal of Water Resources Development, 1-26,

https://doi.org/10.1080/07900627.2020.1745159, 2020.

Anghileri, D., Monhart, S., Zhou, C., et al. The Value of Subseasonal Hydrometeorological Forecasts to Hydropower Operations: How Much Does Preprocessing Matter? Water Resources Research, 55(12), 10159-10178, https://doi.org/10.1029/2019WR025280, 2019.

Anghileri, D., Voisin, N., Castelletti, A., et al. Value of long-term streamflow forecasts to reservoir operations for water supply in snow-dominated river catchments. Water Resources Research, 52(6), 4209-4225, https://doi.org/10.1002/2015WR017864, 2016.

Feng, D., Fang, K., Shen, C. Enhancing streamflow forecast and extracting insights using long-short term memory networks with data integration at continental scales. Water Resources Research, 56(9), e2019WR026793, https://doi.org/10.1029/2019WR026793, 2020.

Gibbs, M. S., McInerney, D., Humphrey, G., et al. State updating and calibration period selection to improve dynamic monthly streamflow forecasts for an environmental flow management application. Hydrology and Earth System Sciences, 22(1), 871-887, https://doi.org/10.5194/hess-22-871-2018, 2018.

Goddard, L., Aitchellouche, Y., Baethgen, W., et al. Providing Seasonal-to-Interannual Climate Information for Risk Management and Decision-making. Procedia Environmental Sciences, 1, 81-101, https://doi.org/10.1016/j.proenv.2010.09.007, 2010.

Hadi, S. J., Tombul, M., Salih, S. Q., et al. The capacity of the hybridizing wavelet transformation approach with data-driven models for modeling monthly-scale streamflow. IEEE Access, 8, 101993-102006, https://doi.org/10.1109/ACCESS.2020.2998437, 2020.

Nanda, T., Sahoo, B., Chatterjee, C. Enhancing real-time streamflow forecasts with wavelet-neural network based error-updating schemes and ECMWF meteorological predictions in Variable Infiltration Capacity model. Journal of Hydrology, 575, 890-910, https://doi.org/10.1016/j.jhydrol.2019.05.051, 2019.

Pechlivanidis, I., Crochemore, L., Rosberg, J., et al. What are the key drivers controlling the quality of seasonal streamflow forecasts? Water Resources Research, 56(6), e2019WR026987, https://doi.org/10.1029/2019WR026987, 2020.

Shamir, E. The value and skill of seasonal forecasts for water resources management in the Upper Santa Cruz River basin, southern Arizona. Journal of Arid Environments, 137, 35-45, https://doi.org/10.1016/j.jaridenv.2016.10.011, 2017.

Sharma, S., Siddique, R., Reed, S., et al. Hydrological Model Diversity Enhances Streamflow Forecast Skill at Short-to Medium-Range Timescales. Water Resources Research, 55(2), 1510-1530, https://doi.org/10.1029/2018WR023197, 2019.

Van Osnabrugge, B., Uijlenhoet, R., Weerts, A. Contribution of potential evaporation forecasts to 10-day streamflow forecast skill for the Rhine River. Hydrology and Earth System

Sciences, 23(3), 1453-1467, https://doi.org/10.5194/hess-23-1453-2019, 2019.

Xu, W., Zhang, C., Peng, Y., et al. A two stage Bayesian stochastic optimization model for cascaded hydropower systems considering varying uncertainty of flow forecasts. Water Resources Research, 50(12), 9267-9286, https://doi.org/10.1002/2013WR015181, 2014.

**Comment #2:** It is unclear how the weight matrices involved in the forecasting models (Lines: 124 and 136) were estimated, and what / which criteria were used in calibration.

**Authors' response:** Thanks for your comment. We will modify it in the new version. "*Both LSTM and GRU are trained based on truncated Back Propagation Through Time (BPTT) which uses a back propagation network to update the parameters in iterations (Cheng et.al., 2020). The NSE function is used as the loss function to calibrate the LSTM and GRU models.*"

**References**

Cheng, M., Fang, F., Kinouchi, T., et al., 2020. Long lead-time daily and monthly streamflow forecasting using machine learning methods. Journal of Hydrology, 590, 125376.

**Comment #3:.** It is left unexplained:

• How the parameters used to define the operational policy are estimated?

**Authors' response:** Thanks for your comments. The parameters in the operation policy are the decision variables in our multi-objective problem, and can be estimated by NSGA-II.

• What specific hydrological variables are included in the "policy inputs"?

**Authors' response:** The hydrological variables in the policy inputs include fore-bay water level, observed or predicted inflows, and precipitation.

• How these "policy inputs" are related to the decision horizon?

**Authors' response:** Thank you. As show in Eq (32), in each operation horizon, $\Gamma_t$ is the $t^{\text{th}}$ policy inputs including exogenous information (e.g., fore-bay water level observed or predicted inflows and precipitation)

$$u_t^k = \sum_{i=1}^{N} \omega_{i,k} \varphi_{i,k}(\Gamma_t),$$ (32)

- How the policy could be implemented with all constraints enforced in a day-by-day practice?

**Authors' response:** Thank you. When using the parameterized MORDM approach to solve the multi-objective reservoir operation under uncertainty, it is indeed hard to obtain the policy that is subject to with all constraints. To avoid this potential problem, we have applied a post-processing procedure in the practice. For example, assume that $Q_{t,i,j}^{r,2}$ denotes the flow pumped by the $j$th pump station from the $i$th reservoir at $t$th time step (m³/s); $Q_{t,j}^{p}$ denotes flow through the $j$th pump station at $t$th time step (m³/s), $Q_{t,j}^{p} = \sum_{i}^{N_j} Q_{t,i,j}^{r,2}$, $N_j$ is the number of reservoirs pumped by the $j^{th}$ pump station; $Q_{j,\max}^{p}$ denotes the upper flow boundary of the $j^{th}$ pump station (m³/s). In some cases, $Q_{t,j}^{p}$ can be greater than $Q_{j,\max}^{p}$, and we will do the following step $Q_{t,n,j}^{r,2} {}' = \dfrac{Q_{t,n,j}^{r,2}}{\sum\limits_{n=1}^{N_j} Q_{t,n,j}^{r,2}} \times Q_{j,\max}^{p}$ to update $Q_{t,n,j}^{r,2}$. The post-processing procedure will be described in Part "3.2 Problem formulation.

- Why it is called "multi-objective" since involving only an objective (26)?

**Authors' response:** Thanks for your constructive comment. In this study, we focus on the multi-objective problem, and three objectives are considered in our case study. Accordingly, it should be multi-objective in this equation, and we will modify it as:

$$p_\theta^* = \arg\min_{p_\theta} \left( J_1, J_2, \ldots, J_M \right)_{p_\theta} \quad s.t. \ \theta \in \Theta,$$ (31)

where $J_1, J_2, \ldots, J_M$ is the objective function, and $M$ is the number of objectives.

Moreover, to answer these above questions, we will re-organize the introduction of the Parameterized multi-objective robust decision making (MORDM).

*"2.4 Parameterized multi-objective robust decision making (MORDM)*

[revised manuscript text omitted]

**Comment #4:** I think this work has formulated an incomplete reservoir operation problem.

- The water balance, for instance, does not reflect the hydraulic connections shown in Figure 4. The relationships between water supply, pumping flow, inflow and discharge are not incorporated in the model.

**Authors' response:** Thanks for your constructive comments. All plants are supplied by the reservoirs, and we can find in Fig.4 that some reservoirs supply water without pump stations (e.g., Longtan, Changchunling, Chahe, and Nanao), while the others will be pumped by pump stations. Assume that $Q_{t,i}^{r}$ denotes flow from the $i$th reservoir at $t$th time step (m$^3$/s), in which $Q_{t,i}^{r,1}$ denotes the flow without pump station from the $i$th

reservoir at $t^{th}$ time step (m³/s), $Q_{t,i,j}^{r,2}$ denotes the flow pumped by the $j^{th}$ pump station from the $i^{th}$ reservoir at $t^{th}$ time step (m³/s). $W_t^s$ denotes the amount of water supply for plants at $t^{th}$ time step (m³), $W_t^s = \sum_{i=1}^{I} \sum_{t=1}^{T} Q_{t,i}^r \Delta t$, $I$ is the total number of reservoirs; $Q_{t,j}^p$ denotes water through the $j^{th}$ pump station at $t^{th}$ time step (m³/s), $Q_{t,j}^p = \sum_{i=1}^{N_j} Q_{t,i,j}^{r,2}$, $N_j$ is the number of reservoirs pumped by the $j^{th}$ pump station. The relationship between water supply and discharge, and that between water supply and pumping flow, are present in the description of Eqs. (38)-(39). The water balance limitation $V_{t+1,i} = V_{t,i} + (I_{t,i} - Q_{t,i}^r) \Delta t$ is mainly for reservoirs. Accordingly, we will modify the problem formulation as bellows.

*"These objective functions are given as follows:*

[revised manuscript text omitted]

$Q_{j,\max}^p$ *, and we will do the following step* $Q_{t,n,j}^{r,2}{}' = \dfrac{Q_{t,n,j}^{r,2}}{\sum\limits_{n=1}^{N_j} Q_{t,n,j}^{r,2}} \times Q_{j,\max}^p$ *to update* $Q_{t,n,j}^{r,2}$ *.*

• Also, how the MORDM is related to this operational problem?

**Authors' response:** Thanks for your comments, we will re-organize the introduction of the parameterized MORDM approach and will describe the detailed steps how MORDM is related to the operational problem.

"*In our study, the parameterized MORDM approach will be coupled with a rolling horizon scheme over one year period to solve the multi-objective reservoir operation problem. Given the lead time of 7 days (forecast horizon is equal to operation horizon) as an example, it is operated following two steps: the optimization model is first operated daily over a 7-day horizon using the parameterized MORDM; after implementing current water allocation decisions, the status, inflow, and other information of reservoirs update as time evolves, and then the remainder is subsequently operated. The two steps are repeated until the process (one year period) is completed. In each operating horizon, the main steps of the parameterized MORDM are: (1) problem formulation, including the possible actions (i.e., RBF parameters) and performance measures; (2) generate alternative RBF policies subjecting to all the constraints and the objectives are evaluated over stochastic inflows; (3) identify solutions with a robust rule (e.g., the principle of insufficient reason, minimax, and minimax regret) using multi-objective evolutionary algorithms (MOEAs) (Giuliani and Castelletti, 2016; Guo et al., 2020b).*"


• The model looks like a linear programming problem that can be easily solved.

**Authors' response:** Thanks for your comments. There are 25 reservoirs and 16 pump stations in our multi-objective reservoir operation optimization problem. Although the objectives and constraints seem to be linear, there are some non-linear functions considered in our modelling process. For example, the relationship between the fore-bay water level and volume of reservoirs is non-linear, and normally expressed by a quadratic function. Besides, it is difficult and time consuming to assure all constraints enforced in the day-by-day practice, especially when it is operated under stochastic inflow.

**Comment #5:** The manuscript will benefit from more logically organizing its contents. The "Results and Discussion" are usually a part of the case studies.

**Authors' response:** Thank you. We will re-organize the manuscript, and put the "3.4 Results and discussion" as a part of case study, and add a part of "3.3 Model development".

Theory, models, procedures and definitions are generally presented before case studies, and some of them need more detailed introduction, including:

• How the weights in the BMA are determined (Line 320)?

**Authors' response:** Thank you. We will modify it as bellows.

*"In this study, a log-like hood function is maximized to estimate the parameters (weight $w_k$ and variance $\sigma_k^2$) as shown in Eq (21).*

$$l(\theta) = \log\left(\sum_{k=1}^{K}\left(w_k \cdot g\left(Q\big| f_k^t, \sigma_k^2\right)\right)\right), \qquad (21)$$

*where $\theta$ is the vector of parameters $\left\{w_k, \sigma_k^2, k = 1, 2, ..., K\right\}$.*

*The Expectation-Maximization (EM) algorithm is used to find out the maximum likehood with a termination criterion (early stopping or a maximal iteration). As the EM proceeds, the parameters of weight $w_k$ and variance $\sigma_k^2$ are updated as follows.*

$$w_k^{(Iter)} = \frac{1}{NT}\left(\sum_{t=1}^{NT} z_k^{t^{(Iter)}}\right), \tag{22}$$

$$\sigma_k^{2^{(Iter)}} = \frac{\sum_{t=1}^{NT} z_k^{t^{Iter}} \cdot \left(Y^t - f_k^t\right)^2}{\sum_{t=1}^{NT} z_k^{t^{Iter}}}, \tag{23}$$

$$z_k^{t^{(Iter)}} = \frac{g\left(Q\middle|f_k^t, \sigma_k^{2^{(Iter-1)}}\right)}{\sum_{k=1}^{K} g\left(Q\middle|f_k^t, \sigma_k^{2^{(Iter-1)}}\right)}, \tag{24}$$

$$l(\theta)^{(Iter)} = \sum_{t=1}^{NT} \log\left(\sum_{k=1}^{K}\left(w_k^{(Iter)} \cdot g\left(Q\middle|f_k^t, \sigma_k^{2^{(Iter)}}\right)\right)\right), \tag{25}$$

*where Iter is the number of iterations. $NT$ is the length of calibration periods. $Y^t$ and*

*$f_k^t$ are the observed and forecast streamflow at $t^{th}$ time step, respectively ($m^3$/s), $z_k^{t^{(Iter)}}$*

*is the latent variable for the $k^{th}$ model at $t^{th}$ time in the Iter iteration.*

• How the Monte Carlo simulation method is used to generate BMA ensemble forecasts (Line 359)?

**Authors' response:** Thank you. We will modify it as bellows.

*"We use the Monte Carlo simulation method to generate BMA ensemble forecasts. The procedure will be described as bellows.*

*a) Generate an integer value of i in [1, 2, ..., K] by using the corresponding probabilities [$u_1$, $u_2$, ..., $u_K$]. Set the initial cumulative weight $w_0^* = 0$ and calculate cumulative weight $w_i^* = w_{i-1}^* + w_i$ for i=1,2,...,K. Create a random variable u between 0 and 1. If $w_{i-1}^* \leq u \leq w_{i-1}^*$, it indicates that the $i^{th}$ model forecast would be selected and used in the next step.*

*b) Generate a realization of the observation $y_t$ using the PDF $g\left(y_t\middle|f_k^t, \sigma_k^2\right)$.*

*c) Repeat the above two steps (a) & (b) for M times. M is the number of Monte Carlo simulation and set as 1000 in this study. Furthermore, 90% confidence intervals between the 5% and 95% quantities were employed to reveal the uncertainty of BMA ensemble forecasts.*

• What "the previous water levels" is supposed to mean (Line 381)?

**Authors' response:** Thanks for your comments. The previous water levels is termed as initial fore-bay water level of reservoirs. We will modify it in the new version.

• Why the NSGA-II are still needed since we already have the operation policy determined (Line 383)?

**Authors' response:** Thanks for your comments. The parameters in the operation policy are the decision variable in our multi-objective problem, and can be estimated by NSGA-II. We will modify it to avoid confusion as bellows.

"*In particular, when DPS problems involve multiple objectives, they can be coupled with truly multiobjective optimization methods, such as multiobjective evolutionary algorithms (MOEAs), which allow estimating an approximation of the Pareto front in a single run of the algorithm (Giuliani et.al., 2016).*"

"*The optimization is solved at each time step (a particular forecast horizon, e.g., 1-7 days) by applying NSGA-II to search the space of decision variables and identify the islands' water allocation trajectories.*"

**References**

Giuliani, M., Castelletti, A., Pianosi, F., et al. Curses, tradeoffs, and scalable management: Advancing evolutionary multiobjective direct policy search to improve water reservoir operations. Journal of Water Resources Planning and Management, 142(2), 04015050, https://doi.org/10.5334/jors.293, 2016.

• How the deterministic, uncertain and observed streamflow are used in the operation (399)?

**Authors' response:** Thank you. We will modify it in the part "3.3 Model development" as bellows.

"*In this study, we use the parametrized MORDM approach to design operating policies for the multi-objective reservoir operations under uncertainty. The optimized operations are both regulated based on deterministic and uncertain forecast inflow. To keep fair, we perform a simulation to generate deterministic and observed ensemble*"

*forecasts that each deterministic and observed data are repeated 900 times, respectively. Using the uncertain streamflow forecasts (BMA, deterministic or observed ensemble forecasts) as policy inputs in the parametrized MORDM method, we can generate alternative RBF policies subjecting to all the constraints and the objectives are evaluated over stochastic inflows."*

• How the Pareto solutions are identified (Line 387)?

**Authors' response:** Thanks for your comments. We do not identify the Pareto solutions. In this study, we focus on assessing the overall operating performance of the multi-reservoir system under different streamflow forecast configurations (i.e., deterministic or stochastic). Accordingly, instead of evaluating the performance of each operation solution, the system operating performances are averaged over the Pareto solutions.

• Whether or not the annual revenues, costs, and water supply reliability (Line 409) are used as multiple objectives when determining the operating policy?

**Authors' response:** Thanks for your comments. We do not use the annual revenues, costs, and water supply reliability as the objectives. We deal with a real-time optimization problem in our study, and assume that the operating policy is determined by the stochastic short-term reservoir inflow forecasts. Accordingly, the indictors of revenues and water supply reliability over the corresponding short-term operating period are termed as the objectives. The annual revenues, costs, and water supply reliability, are just chosen as metrics to compare and assess the performance of the operating policies derived from different configurations.

• "Fake" results do not have any meaningful value, so why they are included in Table 6 in the first place (Line 428)?

**Authors' response:** Thank you. To the best of our knowledge, there are few inform-driven studies have clearly point out that whether or not the system operating performance is post-evaluated by the true streamflow information. The differences between Table 6 and Table 7 may provide references for beginners.

**Comment #6:** To the best of my understanding, the NSE was used to calibrate the forecasting models while the RMSE and MAE are also used in assessing the

performance of the models. I think it should be a fairer practice by using multi-criteria to do both the calibration and assessment.

**Authors' response:** Thank you. Indeed, it is fairer by using multi-criteria to do both the calibration and assessment. However, in our study, we aim to identify the relationship between forecast skill and forecast-driven reservoir operation. To answer this question, five input combination scenarios are investigated and two of them are then applied to drive the multi-objective reservoir operation optimization. Accordingly, we prefer to distinguish the forecast skill of different configurations using the indictors of NSE, RMSE, and MAE, rather than improving the forecast skill. But it may be interesting to obtain forecasts when accounting for multi-criteria over both calibration and assessment period. We will add some discussion in the Part "Limitation and future work" as "*Our work suffers from some limitations, which could be overcome in future studies. One of the limitations is that the single indictor is used to calibrate the forecast models while multiple indictors are used in assessing the performance of the models. It should be a fairer practice by using multi-criteria to do both the calibration and assessment and can be interesting as a future work.*"

**Comment #7:** Please justify why the Radial Basis Functions are used to parameterize the policy (Line 199)?

**Authors' response:** Thank you for your suggestion. We will modify it. "*Different DPS approaches have been proposed, where two nonlinear approximating networks, namely artificial neural networks (ANNs) and radial basis functions (RBFs) have become widely adopted as universal approximators in many applications (Deisenroth et al., 2013; Giuliani et al., 2016). In particular, we parameterize the operating policy as RBFs, because they have been demonstrated to be effective in solving multi-objective water resources management problems (Giuliani et al., 2014; 2015).*"

**Comment #8:** Including the test period when minimizing the NSE (Line 285) will make it lose efficacy in assessing the model performance in future.

**Authors' response:** Sorry for the confusion. This should be "*As for LSSVM, we avoid overfitting by minimizing the NSE during the calibration and validation periods, while the test period is also used to assess the forecast performance.*"

**Technical Corrections:**

**Comment #1:.** Please rewrite the term ($\sum_{i=1}^{K} w_k f_k$) in equation (19), which just does not make sense to me, with the $f_k$ being a model.

**Authors' response:** Thanks for your comments. We will revise "model $f_k$" as "model forecast $f_k$".

**Comment #2:** Please double check all the mathematical expressions.

- In equations (19) and (20), the sum should be operated over subscript "$k$" rather than "$i$".

**Authors' response:** Thank you. We will change "$i$" to "$k$" as bellows.

$$E\left[Q|D\right] = \sum_{i=1}^{K} w_k \cdot E\left[p_k\left(Q|f_k, D\right)\right] = \sum_{i=1}^{K} w_k f_k \tag{2}$$

$$V\left[Q|D\right] = \sum_{i=1}^{K} w_k \cdot \left[f_k - \sum_{i=1}^{K} w_k f_k\right]^2 + \sum_{i=1}^{K} w_k \sigma_k^2 \tag{3}$$

• It might not be right that the subscript "k" on the left side does not appear on the right side of the equation (28)

**Authors' response:** Thank you. We will revise it as bellows.

$$\varphi_{i,k}(\Gamma_t) = \exp\left[-\sum_{j=1}^{M} \frac{\left[(\Gamma_t)_j - c_{j,i,k}\right]^2}{b_{j,i,k}^2}\right], \tag{4}$$

• It sounds not right to me in equation (34), where a variable without subscript "*j*" is summed over "*j*".

• It is questionable that the equation (35) does not have a subscript for the first sum operator to operate over.

• Expressing a variable subscript "*n*" in "$Q_{max\_n}$" (Line 247) is something strange.

• Please check on all similar unprofessional expressions in (39), (42) and (43).

**Authors' response:** Thank you. We will modify these equations as bellows.

"

[revised manuscript text omitted]

(2) Reservoir storage limits: $\qquad\qquad V_{t,i,\min} \leq V_{t,i} \leq V_{t,i,\max}$, $\qquad$ (47)

(3) Reservoir release limits (for the reservoir that supply water without pump station): $\qquad\qquad Q_{t,i}^r \leq Q_{t,i,\max}^r$, $\qquad$ (48)

(4) Pumping station limits: $\qquad\qquad Q_{t,j}^p \leq Q_{\max,j}^p$, $\qquad$ (49)

where $I_{t,i}$ is the inflow of the $i^{th}$ reservoir at $t^{th}$ time step (m³/s); $V_{t,i}$ is the storage of $i^{th}$ reservoir at $t^{th}$ time step (m³); $V_{min}$ and $V_{max}$ are the lower and upper storage boundaries, respectively (m³); $Q_{t,i,\max}^r$ is the maximum release of the $i^{th}$ reservoir at $t^{th}$

*time step (m³/s). In some cases,* $Q_{t,j}^{p}$ *obtained by the RBF policies can be greater than*

$Q_{j,\max}^{p}$ *, and we will do the following step* $Q_{t,n,j}^{r,2}{}' = \dfrac{Q_{t,n,j}^{r,2}}{\displaystyle\sum_{n=1}^{N_j} Q_{t,n,j}^{r,2}} \times Q_{j,\max}^{p}$ *to update* $Q_{t,n,j}^{r,2}$ *."*

**Comment #3:** Please do not omit subscripts in mathematical symbols. And for all the definitions of math symbols, all the subscripts in any symbol should appear in its definition.

**Authors' response:** Thank you. We will double check all the subscripts in mathematical symbols.

---

## Author Comment (AC2) · 25 Mar 2021

Dear Referee #2,

We highly appreciate your review and useful comments for our manuscript. We provide our answers to your queries below.

Kind regards, all authors

Queries by anonymous referee #2 RC2 & answers by authors are as follows:

**Comment #1:** deterministic S5 performance indicators overlap each other, I would suggest to modify color or width to improve the readability.

**Authors' response:** Thanks for your constructive comments. We will revise this figure as bellows.

[Figure]

**Figure 6: NSE values at lead times of 1 to 7 days plotted against the coefficient of variation (COV) for all the 24 reservoirs during the period of (a) calibration, (b) validation, and (c) test under S5.**

---

## Editor Comment (EC1) · Dimitri Solomatine (Editor) · 5 Apr 2021

Dear authors, The referees provided quite useful comments. They are also asking to provide better formulations, and to work on improving the structure of the paper. In terms of English, it is always useful to ask help of a professional proof-reading company. Your answers convince me that you know the way forward, and it will not take too much time to prepare the revised manuscript. Success with the revision!

I am aslo attaching a piece, which you may find useful when wotking on the paper revision.

Please also note the supplement to this comment:
https://hess.copernicus.org/preprints/hess-2020-617/hess-2020-617-EC1-
supplement.pdf

───────────────────────────────

**Supplement:**

**Preparing a paper and dealing with reviews:**
**how to increase chances for your paper to be published**

**D.P. Solomatine**

*These are some suggestions that may help the authors in paper preparation. It refers mainly to papers based on (mathematical, computer) modelling and optimization studies tested on a case study, but could be useful to the authors of other types of papers submitted to various journals as well.*

**PREPARING A PAPER**

How to clearly and logically present your research results in the form of a scientific paper with an impact? You think the research you have carried is sound and it carries enough innovation, but how to convince others? A paper may have various forms, and researchers are free to express their ideas as they wish. However, it is often useful to follow generally accepted ideas of what text is seen to be logical, convincing and well written, and then chances that it is accepted by a renowned peer-reviewed journal are higher.

You are submitting a paper to a scientific journal, and it will be reviewed by researchers who know this subject quite well. Your paper have to present enough innovation, and present it clearly and convincingly. Typically they tend to reject papers presenting just another example of applying a known method to yet another typical case study. (If you do not see much of scientific innovation, but still think there are some new interesting elements in your work which can be useful for professionals, you may want to consider submitting a paper as a "technical note" or "experience paper" to a professional journal.)

A scientific paper typically follows the following logical structure:
- abstract (should briefly present motivation, objectives, novelty, and main finding(s));
- motivation for this work (introduction) (background; what is the problem? why it is important to solve it? what was already done before and why more research is needed?). This section is often called Introduction or Motivation;
- objectives (main objective, possibly with specific objectives);
- methodology (methods) with the justification of their use;
- description of the case study, with the specific problems associated with it, and data availability;
- experimental setup (details of data sets, parameterisation of models);
- results and discussion;
- conclusions: main findings, limitations, future outlook.

Titles could be a bit different (like Materials and Methods), and the order may vary (e.g. case study is sometimes presented in Introduction).

**Motivation (Introduction)** presents the background of this research work. Present the phenomenon or a system (physical, biological, socio-economical, etc.) you are researching, and the research problem which is to be solved. Explain why it is important to do this research (solve this problem), and if appropriate present a wider context of the problem, e.g. the social,

environmental and other aspects. Present the previous examples of addressing the problem, present a critical review of the known literature (stating what was positive and interesting, what were the conclusions, and what was not sufficiently researched in your opinion). Avoid presenting the known "text-book" material, descriptions of widely used models, etc. - provide references. Based on literature review, justify why more (your) research is needed.

Formulating the problem, try to highlight the starting point and to attribute it to one of the following classes: a) "case study first": to solve some problems related to the case study (which lead to introduction of new methods/models), or b) "method first": you developed an interesting method and want to test it on a case study. (Often however these two types are mixed.)

Based on the problem formulation, formulate the objective. It can be also detailed into several specific objectives, and/or research questions.

Clearly present what is the innovation of your study, how does it enrich science and/or technology. What is new in your research? Mention what is its practical value.

You may also provide a brief outline of the paper.

**Methodology**. Try to present a diagram or logical scheme of the main parts of methodology logically arranged. Describe the important methods used in this paper, so that a reader would understand the main principles without going to other literature sources. Explain what are the assumptions made about data, its accuracy and stationarity, and about the adequacy of the models used.

Logic of introducing (using) any (new) approach/method/model is this: 1) show the deficiencies of an existing method(s); 2) introduce the new one and justify its choice; 3) compare the new one to the old one(s) and demonstrate the advantages and limitations of the new one.

If you use a complex model A, explain why a simpler model B cannot be used, and/or in what sense the use of a more complex model would help. Example: if you suggest to use an ANN, show the deficiencies of a simpler model, e.g. of a linear regression model. If a linear model is good enough, there is not much use of developing a non-linear one.

Present the model performance indicators to be used for assessing models accuracy (comparing its output to observed data).

Modelling methodology typically includes: data analysis, data cleaning and infilling, sometimes data transformation, model calibration, analysis of model sensitivity and uncertainty to its parameters and inputs (and sometimes to model structure), interpretation of results. If for some reason you do not calibrate a model, or do not intent to apply formal methods for uncertainty analysis, explain why, and try to give some ideas of how it can be done.

If you are solving an optimization problem, this problem has to be presented mathematically, with the three explicit components: decision variables, constraints, and objectives functions. If you use randomised search methods (like evolutionary algorithms) explain, why (faster) gradient-based methods cannot be used.

**Case study**. Describe the case study, with the specific problems associated with it. For a hydrological or hydraulic case study, description of the physical system (e.g. catchment, or a river) and a map is often required. Describe what is data is available, how accurate is it.

**Experimental setup**. The details of your experiments, e.g. details of data sets, parameterisation of models, the used hardware, etc. is good to put in this separate section. Methodology is quite a general thing, and the details of experiment are specific for the considered case study or models tuning. This section does not present the results, but only the "setting the scene".

If a data-driven or statistical model is presented, the following is expected:
- Description of data, plots, etc. Try to answer the question: is the available data size and quality enough to build a reliable data-driven model?
- Explanation of how input variables are chosen. Typically correlation or mutual information analysis is used, or model-based methods (i.e. a set of variables is chosen ensuring the maximum performance on cross-validation set). Of course expert judgement is also one of the methods to use, but its validity and underlining reasoning has to be presented.
- Is there a match between the amount of data available and the number of parameters in a model to determine (model structure). For example, 30 examples is hardly enough to train a MLP ANN with 120 weights.
- Explanation on how data is split in 1) training set, 2) cross-validation set, 3) test set. (Sometimes there are only training and test sets).
- It is methodologically incorrect to use the test set (designed to imitate functioning of the model in operation) for tuning the model, or selecting the best model. Only training and cross-validation sets can be used for that.
- Presentation of the statistical properties of the data (if applicable): of the whole set, and of the mentioned sub-sets: min, max, mean, standard deviation, size.
- Explanation on how cross-validation is carried out? (ideally, try to apply n-fold cross validation). If cross-validation is not employed please explain why.

Sometimes the usefulness of a method (model) is demonstrated on a synthetic data set (because you can easily control its properties, noise in data, etc. to explore behaviour and performance of a model), and then it is done for a real-life case study.

**Results and discussions**. In this section the results are presented. Use tables and graphs to make it convincing. The experiments have to show e.g. that the method you propose, in comparison with the methods used in the past, is better or at least not inferior, based on some criteria: accuracy, speed, generality, ease of use, etc. It may happen that your model shows very similar results - in this case think of the scenarios of using it: it may be used for cross-comparison, or in ensemble with other methods, etc.

This section does end with presenting the results, e.g. plots comparing various methods. The results have to interpreted, you have to try to explain unexpected results which may contradict common sense. Try to link the results of computer model experiments with reality, the ways of communicating these results to practitioners and the use of these methods or models for other case studies. Discuss how strong were the assumptions you made (about data and models), and their influence on the results.

Discuss the results of model sensitivity and uncertainty analysis.

When comparing two methods of models A and B, under assumption of noisy data, please take into account the following. If the differences in their performance are small (several percent), be careful in making strong statements like "model A is better than model B". These differences may disappear if you split data differently, or parameterise your model differently. It is better to use words like "slightly better, comparable, marginally better, or has similar performance". If you know data is not very accurate (noisy), there is no sense of using too many significant digits in the presentation of errors.

Discuss how the method or model you present will be used in practice, what particular practical problems will it help to solve (if applicable), how to use this methods in other cases and contexts.

Present and discuss the limitations of your approach. Discuss scenarios when your approach may not work.

**Conclusions and recommendations**. In the section with conclusions the following can be mentioned. Remind readers what they have read, why it was significant and why this paper was worth reading and publishing. Main findings (good if they are numbered - this shows your structured thinking). An answer to the question: are objectives reached (could be even done for each specific objective). Limitations of the presented approach. Can the results be generalised? (or strong assumptions do not allow for that). How these results may change professional practice, what are the recommendations of using the presented methodology in other cases? Briefly explain what are the directions for further research (future outlook).

Sometimes it is useful to put the "take home message" or mention the "iconic figure" of your paper.

It is better to be modest in stating what you have achieved, and use words like "based on the two experiments, the presented combination of methods has certain advantages if compared to other methods applied earlier", "it can be concluded that the presented model can help management practice in the given case study", rather than "the model is superior", "method A is better than method B". (The latter can be indeed true - but only for the considered cases.)

**Figures**: Avoid excessive numbers of figures: select those figures that clearly support the presentation. Provide an explaining caption allowing the reader to understand the figure without reading the main text. Quality: are axes marked, are legends clear, and captions self-explanatory, are plots in black-and-white print clear? In the original submission embed figures within main text to avoid reviewers needing to move back and forth.

**References**: Have you provided the latest references? (A good check is to use Internet search engines to find papers with similar keywords.) Are they properly formatted? If you submit the paper to journal J, it would be reasonable to provide references to the papers in this particular journal. Please note that it is not very good taste to have too many references to your own papers. Do all citations in the text have a corresponding reference? Are all the references cited in the text?

**Plagiarism**. This is an important issue - there is enough material in Internet on this topic. If you are using ideas or material form another paper, always provide a citation. If you take more than seven words in a row, enclose them in quotation marks. Every figure or drawing has an author. If you want to present a figure from another source you may need a permission of the publisher. Please note that direct use of text or Figures from your own earlier publication is not allowed by a publisher. This may present problems if you want to use e.g. a figure with a model structure in several publications. (What some authors do is the following: they make figures and drawings for their research and put them on their web page; this material is considered to be public, cannot be copyrighted, and can be used in the papers submitted to journal by everybody including the author - of course with the corresponding reference.)

**Style**: please try to avoid repeating the same thing several times trying "to convince the reader". For each paragraph, try to answer the following questions: what exactly do I want to say here? is this absolutely clear for the reader who is seeing this paper for the first time? Avoid long sentences and making multiple statements in one sentence. Have one paragraph for each distinct topic. Try to provide a logical transition from one section (paragraph) to another to ensure a clear flow of thought, guiding the reader from one topic to another.

Check if you have more than one sub-sections in every section (or none). If a section has several sub-sections, ensure that every considerable part of text has a title and constitutes a sub-section.

Be consistent with time used: if you started e.g. to use Present Indefinite Tense ("our experiments show") (this is perhaps the best choice), use it across the paper and don't switch to Past Indefinite ("our experiments showed") or Present Perfect ("our experiments have shown"). If you present earlier results of other researchers, the use of Present Perfect Tense is perhaps the most appropriate ("authors have shown").

**English**: is it of the level that you see in the published papers? Try to polish English as much as possible before submission. Try to go through every sentence and check if it has sense and is **clearly formulated** (help from a colleague could be a thing to consider).

**What reviewers to recommend?** Usually the authors are asked to give names of several potential reviewers. Please try ensure that they are independent, not from you Institution, come from several countries, and you may want even to think of various nationalities. Try not to recommend close personal friends since you may be putting them in a difficult position. If you are providing a reviewer's name, and his/her papers are not cited in your paper, please think what expertise you assume this reviewer does have that you are recommending this person. Please do not recommend reviewers who approve anything - you will miss the chance to improve the paper, the editor will have to search for the new reviewers, and this will delay publication. Try to think of reviewers as people who are experts, may give a critical view at your paper, point at possible problems, give recommendations, and by this may help to improve it before publication.

**WHAT THE REVIEWERS ARE LOOKING AT: A CHECKLIST**

Before submission, try to put yourself into the shoes of a reviewer, and think what a reviewer would be analysing in your paper. Read the paper as if you are a reviewer.

When evaluating your paper, the reviewers will typically try to answer the following questions.

Is the work understandable, and appropriate for the journal?
- Is the purpose or goal of the work within the journal's scope?
- What are the problems to solve in the paper? Are they clearly stated?
- Are the techniques employed appropriate? Is the mathematics correct?

Is the work original and interesting?
- Does this work contain new results that significantly advance the research field?
- Have any parts of the paper already been published or considered for other publication?

Is this paper likely to be cited in the future?
- Is the paper scientifically sound? Does it provide sufficient information and in-depth discussion?
- Are the results clearly and convincingly presented? Can they be reproduced?

Is the presentation logical and clear?
- Does the work follow a traditional logical structure: "motivation - problem statement - objectives - methodology - results and discussion - conclusions - future outlook"?
- Is it enough to read the abstract and understand the main objectives and findings?
- Does the introductory section adequately explain the problems to be solved by this research? Are the importance and usefulness of this research work clear?
- Is the case study, and the problems associated with it, clearly presented?
- Is the conclusion logically supported by the obtained results?
- Are the limitations of the presented approach discussed?
- Is the paper clearly written?
- Does the title reflect the contents of the paper?
- Are sufficient references cited for providing a background to the research?
- Is the length and format of the paper appropriate?
- Are the figures and tables easily readable, correct and informative?
- Is the English understandable? Is the paper free of typographical and grammatical errors?

**HOW TO DEAL WITH THE REVIEWERS' COMMENTS?**

It may happen that your paper is not immediately accepted, and the Editor recommends revision. Read the reviewers' comments, and take a deep sigh. You may be asked to do more experiments, and usually you will have to spend quite a lot of time on writing the reply (sometimes called a rebuttal) to reviewers' or editor's comments. If a rebuttal is well-written, it will help to establish a good positive communication with the editor and reviewers.

In almost all cases reviewers want to help improve the paper. Do not take comments personal, even if sometimes you may not like the way comments are expressed. Note that for the most reviewers English is not the mother tongue and they are busy people.

Please provide answers to every comment. Try to be clear, convincing, and at the same time brief. Please provide citations from the revised manuscript in the reply, so that the reviewer

could see what is changed. It is is also useful to indicate with colour, what is changed in the revised manuscript (or using Track Changes in MS-Word).

Sometimes reviewers pose questions instead of stating what should be changed. It means that perhaps you were not very clear in your manuscript. When you are writing a reply to a reviewer's question, please understand the reader will not see this reply and your explanations. Typically, if such reply is needed, it means that the manuscript should be revised accordingly as well.

Admit your errors, and explain why you have made them. However you may not agree to some comments. In this case, explain with what you do agree, and with what you do not, and why. Sometimes reviewers ask to do more work. It can be a justified suggestion, but you also may think that the paper is already too long, and have enough to present. In this case, if you think you have a strong point, give your reasons why you think it would not be reasonable to implement the reviewer's suggestion.

It may happen that two reviewers recommend contradicting things. Either find a compromise, or choose which way to go. In any case, explain, why you do not follow a particular reviewer's suggestion, mentioning that you have conflicting suggestions, and why have you chosen one of them against the other.

It may sometimes happen that a reviewer is not as knowledgeable on the subject of the manuscript as the author. Some papers may be so innovative that may be rejected simply because reviewers would not appreciate the results. Nonetheless, reviewer comments could be useful, even if they are wrong. Try to be patient, appreciate what the reviewer is saying, try to understand his/her reasoning, and explain clearer in your rebuttal why do you think your results are correct.

It may happen that a reviewer's comment is very brief, unsubstantiated and does not provide enough information. In this case you may write to the associate editor to request another review.

Some reviewers may ask to include references to papers which may not be necessarily relevant. Or they recommend to include more (or less) figures, or extend particular parts. You may find certain comments and requests unreasonable. Take them into account, but it is your decision to what extent to change the manuscript. If you are reluctant to follow some suggestions, be polite and provide a clear and justified explanation why. You are the author, not the reviewers.

Many critical reviewers' comments result from poor writing that leads to a reviewer's misunderstanding. In your reply, admit you were not clear, explain the possible misunderstanding, and show how the manuscript is updated.

Before submitting the revised manuscript, imagine you are a novice reader, and read it slowly again. Is it now clear, logical and convincing? It may be also useful to read yet another time what is written above in this document, and to try to check if you did everything possible to ensure the revised manuscript will be accepted.

Good luck!

---

## Author Response (AR1)

**Reply to the comments on hess-2020-617**

Dear editor and reviewers,

Thank you very much for your evaluation of our manuscript and insightful comments, which have been a great help in improving the quality of our manuscript. We have carefully revised the manuscript according to these comments and suggestions. The related parts of the manuscript have been rewritten and improved, and for your easy reading and evaluation, the changed parts are marked using track changes text in the revised version.

**Reply to the comments from Referee #1,**

**Comment #1:** Some comments in literature review could be more precisely.

- The LSTM and GRU, for example, were not only applied in few previous works (refer to Line 55 in the manuscript)

**Authors' response:** Thank you. We have modified Line 55 in the new manuscript. Please see Lines 52-56, Page 2.

*"LSTM and GRU networks have been successfully applied in many fields (Greff et al., 2017; Zhang et al., 2018; Jung et al., 2020; Shahid et al., 2020; Ayzel and Heistermann, 2021), and they are demonstrated to generate comparable performances. But GRU has a more straightforward structure and a higher operation speed than LSTM. Recently, many applications that assessed them together are also found in the hydrological field (Gao et al., 2020; Muhammad et al., 2020)."*

**References**

Ayzel, G., Heistermann, M. The effect of calibration data length on the performance of a conceptual hydrological model versus LSTM and GRU: A case study for six basins from the CAMELS dataset. Computers & Geosciences, 149, 104708, https://doi.org/10.1016/j.cageo.2021.104708, 2021.

Gao, S., Huang, Y., Zhang, S., et al. Short-term runoff prediction with GRU and LSTM networks without requiring time step optimization during sample generation. Journal of Hydrology, 589, 125188, https://doi.org/10.1016/j.jhydrol.2020.125188, 2020.

Greff, K., Srivastava, R. K., Koutník, J., et al. LSTM: A Search Space Odyssey. IEEE Transactions on Neural Networks and Learning Systems, 28(10), 2222-2232, https://doi.org/10.1109/TNNLS.2016.2582924, 2017.

Jung, Y., Jung, J., Kim, B., et al. Long short-term memory recurrent neural network for modeling temporal patterns in long-term power forecasting for solar PV facilities: Case study of South Korea. Journal of Cleaner Production, 250, 119476, https://doi.org/10.1016/j.jclepro.2019.119476, 2020.

Muhammad A.U., Li X., Feng J. Using LSTM GRU and Hybrid Models for Streamflow Forecasting. Machine Learning and Intelligent Communications 2019. Lecture Notes of the Institute for Computer Sciences, Social Informatics and Telecommunications Engineering, Springer, 294, 510-524, https://doi.org/10.1007/978-3-030-32388-2_44, 2019.

Shahid, F., Zameer, A., Muneeb, M. Predictions for COVID-19 with deep learning models of LSTM, GRU and Bi-LSTM. Chaos, Solitons & Fractals, 140, 110212, https://doi.org/10.1016/j.chaos.2020.110212, 2020.

Zhang, D., Lindholm, G., Ratnaweera, H. Use long short-term memory to enhance Internet of Things for combined sewer overflow monitoring. Journal of Hydrology, 556, 409-418, https://doi.org/10.1016/j.jhydrol.2017.11.018, 2018.

- The research works on impacts of forecast horizon on reservoir operation were not rare (Lines 59 and 71).

**Authors' response:** Thanks for your comments. We have made corresponding revisions in the manuscript, such as:

[revised manuscript text omitted]

**Comment #2:** It is unclear how the weight matrices involved in the forecasting models (Lines: 124 and 136) were estimated, and what / which criteria were used in calibration.

**Authors' response:** Thanks for your comment. We have modified it in the new version. Please see Lines 301-303, Page 14.

*"Both LSTM and GRU are trained based on truncated Back Propagation Through Time (BPTT) which uses a back propagation network to update the parameters in iterations (Cheng et.al., 2020). The NSE function is used as the loss function to calibrate the LSTM and GRU models."*

**References**

Cheng, M., Fang, F., Kinouchi, T., et al., 2020. Long lead-time daily and monthly streamflow forecasting using machine learning methods. Journal of Hydrology, 590, 125376.

**Comment #3:.** It is left unexplained:

• How the parameters used to define the operational policy are estimated?

**Authors' response:** Thanks for your comments. The parameters in the operation policy are the decision variables in our multi-objective problem, and can be estimated by NSGA-II.

• What specific hydrological variables are included in the "policy inputs"?

**Authors' response:** The hydrological variables in the policy inputs include fore-bay

water level, observed or predicted inflows, and precipitation.

• How these "policy inputs" are related to the decision horizon?

**Authors' response:** Thank you. As show in Eq (32), in each operation horizon, $\Gamma_t$ is the $t^{\text{th}}$ policy inputs including exogenous information (e.g., fore-bay water level observed or predicted inflows and precipitation)

$$u_t^k = \sum_{i=1}^{N} \omega_{i,k} \varphi_{i,k}(\Gamma_t), \tag{32}$$

• How the policy could be implemented with all constraints enforced in a day-by-day practice?

**Authors' response:** Thank you. When using the parameterized MORDM approach to solve the multi-objective reservoir operation under uncertainty, it is indeed hard to obtain the policy that is subject to with all constraints. To avoid this potential problem, we have applied a post-processing procedure in the practice. For example, assume that $Q_{t,i,j}^{r,2}$ denotes the flow pumped by the $j^{\text{th}}$ pump station from the $i^{\text{th}}$ reservoir at $t^{\text{th}}$ time step (m³/s); $Q_{t,j}^{p}$ denotes flow through the $j^{\text{th}}$ pump station at $t^{\text{th}}$ time step (m³/s), $Q_{t,j}^{p} = \sum_{i}^{N_j} Q_{t,i,j}^{r,2}$, $N_j$ is the number of reservoirs pumped by the $j^{th}$ pump station; $Q_{j,\max}^{p}$ denotes the upper flow boundary of the $j^{th}$ pump station (m³/s). The post-processing procedure have been described in Part "3.2 Problem formulation. Please see Lines 281-282, Page 13.

*"In some cases, $Q_{t,j}^{p}$ can be greater than $Q_{j,\max}^{p}$, and we will do the following step*

$$Q_{t,n,j}^{r,2}{}' = \frac{Q_{t,n,j}^{r,2}}{\sum_{n=1}^{N_j} Q_{t,n,j}^{r,2}} \times Q_{j,\max}^{p} \quad \text{to update} \quad Q_{t,n,j}^{r,2}."$$

• Why it is called "multi-objective" since involving only an objective (26)?

**Authors' response:** Thanks for your constructive comment. In this study, we focus on the multi-objective problem, and three objectives are considered in our case study. Accordingly, it should be multi-objective in this equation, and we have modified it.

$$p_\theta^* = \arg \min_{p_\theta} (J_1, J_2, ..., J_M)_{p_\theta} \quad s.t. \; \theta \in \Theta, \tag{31}$$

where $J_1, J_2, ..., J_M$ are the objective functions, and $M$ is the number of objectives.

Moreover, to answer these above questions, we have re-organized the introduction of the Parameterized multi-objective robust decision making (MORDM). Please see Lines 194-225, Pages 9-10.

*"**2.4 Parameterized multi-objective robust decision making (MORDM)***

[revised manuscript text omitted]

**Comment #4:** I think this work has formulated an incomplete reservoir operation problem.

- The water balance, for instance, does not reflect the hydraulic connections shown in Figure 4. The relationships between water supply, pumping flow, inflow and discharge are not incorporated in the model.

**Authors' response:** Thanks for your constructive comments. All plants are supplied by the reservoirs, and we can find in Fig.4 that some reservoirs supply water without pump stations (e.g., Longtan, Changchunling, Chahe, and Nanao), while the others will be pumped by pump stations. Assume that $Q_{t,i}^{r}$ denotes flow from the $i^{th}$ reservoir at $t^{th}$ time step (m³/s), in which $Q_{t,i}^{r,1}$ denotes the flow without pump station from the $i^{th}$ reservoir at $t^{th}$ time step (m³/s), $Q_{t,i,j}^{r,2}$ denotes the flow pumped by the $j^{th}$ pump station from the $i^{th}$ reservoir at $t^{th}$ time step (m³/s). $W_{t}^{s}$ denotes the amount of water supply for plants at $t^{th}$ time step (m³), $W_{t}^{s}=\sum_{i=1}^{I}\sum_{t=1}^{T}Q_{t,i}^{r}\Delta t$, $I$ is the total number of reservoirs; $Q_{t,j}^{p}$ denotes water through the $j^{th}$ pump station at $t^{th}$ time step (m³/s), $Q_{t,j}^{p}=\sum_{i=1}^{N_{j}}Q_{t,i,j}^{r,2}$, $N_{j}$ is the number of reservoirs pumped by the $j^{th}$ pump station. The relationship between water supply and discharge, and that between water supply and pumping flow, are present in the description of Eqs. (38)-(39). The water balance limitation $
[revised manuscript text omitted]

*(4) Pumping station limits:* $\qquad Q^{p}_{t,j} \le Q^{p}_{\max, j}$, $\qquad\qquad$ *(49)*

*where $I_{t,i}$ is the inflow of the $i^{\text{th}}$ reservoir at $t^{\text{th}}$ time step (m³/s); $V_{t,i}$ is the storage of $i^{\text{th}}$ reservoir at $t^{\text{th}}$ time step (m³); $V_{min}$ and $V_{max}$ are the lower and upper storage boundaries, respectively (m³); $Q^{r}_{t,i,\max}$ is the maximum release of the $i^{\text{th}}$ reservoir at $t^{\text{th}}$ time step (m³/s). In some cases, $Q^{p}_{t,j}$ obtained by the RBF policies can be greater than $Q^{p}_{j,\max}$, and we will do the following step $Q^{r,2}_{t,n,j}{}' = \dfrac{Q^{r,2}_{t,n,j}}{\sum\limits_{n=1}^{N_j} Q^{r,2}_{t,n,j}} \times Q^{p}_{j,\max}$ to update $Q^{r,2}_{t,n,j}$.*

• Also, how the MORDM is related to this operational problem?

**Authors' response:** Thanks for your comments, we have re-organized the introduction of the parameterized MORDM approach and described the detailed steps in the new version. Please see Lines 216-225, Page 10.

*"In our study, the parameterized MORDM approach will be coupled with a rolling horizon scheme over one year period to solve the multi-objective reservoir operation problem. Given the lead time of 7 days (forecast horizon is equal to operation horizon) as an example, it is operated following two steps: the optimization model is first operated daily over a 7-day horizon using the parameterized MORDM; after implementing current water allocation decisions, the status, inflow, and other information of reservoirs update as time evolves, and then the remainder is subsequently operated. The two steps are repeated until the process (one year period) is completed. In each operating horizon, the main steps of the parameterized MORDM are: (1) problem formulation, including the possible actions (i.e., RBF parameters) and performance measures; (2) generate alternative RBF policies subjecting to all the constraints and the objectives are evaluated over stochastic inflows; (3) identify solutions with a robust rule (e.g., the principle of insufficient reason, minimax, and minimax regret) using multi-objective evolutionary algorithms (MOEAs) (Giuliani and Castelletti, 2016; Guo et al., 2020b)."*


• The model looks like a linear programming problem that can be easily solved.

**Authors' response:** Thanks for your comments. There are 25 reservoirs and 16 pump stations in our multi-objective reservoir operation optimization problem. Although the objectives and constraints seem to be linear, there are some non-linear functions considered in our modelling process. For example, the relationship between the fore-bay water level and volume of reservoirs is non-linear, and normally expressed by a quadratic function; the RBF functions we used are also non-linear. Besides, it is difficult and time consuming to assure all constraints enforced in the day-by-day practice, especially when it is operated under stochastic inflow.

**Comment #5:** The manuscript will benefit from more logically organizing its contents. The "Results and Discussion" are usually a part of the case studies.

**Authors' response:** Thank you. We have re-organized the manuscript and put the "3.4 Results and discussion" as a part of case study and add a part of "3.3 Model development".

Theory, models, procedures and definitions are generally presented before case studies, and some of them need more detailed introduction, including:

• How the weights in the BMA are determined (Line 320)?

**Authors' response:** Thank you. We have modified it. Please see Lines 165-172, Pages 7-8.

*"In this study, a log-like hood function is maximized to estimate the parameters (weight $w_k$ and variance $\sigma_k^2$) as shown in Eq (21).*

$$l(\theta) = \log\left(\sum_{k=1}^{K} \left(w_k \cdot g\left(Q|f_k^t, \sigma_k^2\right)\right)\right), \tag{21}$$

*where $\theta$ is the vector of parameters $\left\{w_k, \sigma_k^2, k = 1, 2, ..., K\right\}$.*

*The Expectation-Maximization (EM) algorithm is used to find out the maximum likehood with a termination criterion (early stopping or a maximal iteration). As the EM proceeds, the parameters of weight $w_k$ and variance $\sigma_k^2$ are updated as follows.*

$$w_k^{(Iter)} = \frac{1}{NT}\left(\sum_{t=1}^{NT} z_k^{t^{(Iter)}}\right), \tag{22}$$

$$\sigma_k^{2^{(Iter)}} = \frac{\sum_{t=1}^{NT} z_k^{t^{Iter}} \cdot \left(Y^t - f_k^t\right)^2}{\sum_{t=1}^{NT} z_k^{t^{Iter}}}, \tag{23}$$

$$z_k^{t^{(Iter)}} = \frac{g\left(Q|f_k^t, \sigma_k^{2^{(Iter-1)}}\right)}{\sum_{k=1}^{K} g\left(Q|f_k^t, \sigma_k^{2^{(Iter-1)}}\right)}, \tag{24}$$

$$l(\theta)^{(Iter)} = \sum_{t=1}^{NT} \log\left(\sum_{k=1}^{K}\left(w_k^{(Iter)} \cdot g\left(Q|f_k^t, \sigma_k^{2^{(Iter)}}\right)\right)\right), \tag{25}$$

*where Iter is the number of iterations. $NT$ is the length of calibration periods. $Y^t$ and $f_k^t$ are the observed and forecast streamflow at $t^{th}$ time step, respectively ($m^3/s$), $z_k^{t^{(Iter)}}$ is the latent variable for the $k^{th}$ model at $t^{th}$ time in the Iter iteration.*

- How the Monte Carlo simulation method is used to generate BMA ensemble forecasts (Line 359)?

**Authors' response:** Thank you. We have modified it as bellows. Please see Lines 172-179, Page8.

*"Then we use the Monte Carlo simulation method to generate BMA ensemble forecasts. Assume M is the number of Monte Carlo simulation and we set M as 1000 in this study. The procedure will be described as bellows.*

*a) Set the initial cumulative weight $w_0^* = 0$ and calculate cumulative weight $w_i^* = w_{i-1}^* + w_i$ for i=1,2,...,K. Create a random variable u between 0 and 1. If*

$w_{i-1}^* \le u \le w_{i-1}^*$, *it indicates that the i*th *model forecast would be selected and used in the*

*next step.*

*b) Generate a realization of the observation y_t using the PDF* $g\left(y_t \mid f_k^t, \sigma_k^2\right)$.

*c) Repeat steps (a) & (b) for M times. Furthermore, 90% confidence intervals between the 5% and 95% quantities were employed to reveal the uncertainty of BMA ensemble forecasts.*

• What "the previous water levels" is supposed to mean (Line 381)?

**Authors' response:** Thanks for your comments. The previous water levels is termed as the initial fore-bay water level of reservoirs. We have modified it in the new version. Please see Lines 317-318, Page 14.

> *"The best operation is obtained by conditioning the operating policies upon the following two input variables, e.g., the initial fore-bay water level and current inflow of reservoir."*

• Why the NSGA-II are still needed since we already have the operation policy determined (Line 383)?

**Authors' response:** Thanks for your comments. The parameters in the operation policy are the decision variables in our multi-objective problem and can be estimated by NSGA-II. We have modified it to avoid confusion. Please see Lines 213-215, Page 10 and Lines 319-320, Page 14.

> *"In general, when DPS problems involve multiple objectives, they can be coupled with truly multiobjective optimization methods, such as multiobjective evolutionary algorithms (MOEAs), which allow an approximation of the Pareto front in a single run of the algorithm (Giuliani et.al., 2016)."*

> *"The optimization is solved at each time step (a particular forecast horizon, e.g., 1-7 days) by applying NSGA-II to search the space of decision variables and identify the islands' water allocation trajectories."*


*"In this study, we use the parametrized MORDM approach to design operating policies for the multi-objective reservoir operations under uncertainty. The optimized operations are both regulated based on deterministic and uncertain forecast inflow. To keep fair, we perform a simulation to generate deterministic and observed ensemble forecasts that each deterministic and observed data are repeated 900 times, respectively. Using the uncertain streamflow forecasts (BMA, deterministic or observed ensemble forecasts) as policy inputs in the parametrized MORDM method, we can generate alternative RBF policies subjecting to all the constraints and the objectives are evaluated over stochastic inflows."*

• How the Pareto solutions are identified (Line 387)?

**Authors' response:** Thanks for your comments. We do not identify the Pareto solutions. In this study, we focus on assessing the overall operating performance of the multi-reservoir system under different streamflow forecast configurations (i.e., deterministic or stochastic). Accordingly, instead of evaluating the performance of each operation solution, the system operating performances are averaged over the Pareto solutions. We have pointed it in the new version. Please see Line 437, Page 18.

*"The system performances are averaged over a set of solutions."*

• Whether or not the annual revenues, costs, and water supply reliability (Line 409) are used as multiple objectives when determining the operating policy?

**Authors' response:** Thanks for your comments. We do not use the annual revenues, costs, and water supply reliability as the objectives. We deal with a real-time optimization problem in our study, and assume that the operating policy is determined

by the stochastic short-term reservoir inflow forecasts. Accordingly, the indictors of revenues and water supply reliability over the corresponding short-term operating period are termed as the objectives. The annual revenues, costs, and water supply reliability, are just chosen as metrics to compare and assess the performance of the operating policies derived from different configurations.

• "Fake" results do not have any meaningful value, so why they are included in Table 6 in the first place (Line 428)?

**Authors' response:** Thank you. To the best of our knowledge, there are few inform-driven studies have clearly point out that whether or not the system operating performance is post-evaluated by the true streamflow information. The differences between Table 6 and Table 7 may provide references for beginners.

**Comment #6:** To the best of my understanding, the NSE was used to calibrate the forecasting models while the RMSE and MAE are also used in assessing the performance of the models. I think it should be a fairer practice by using multi-criteria to do both the calibration and assessment.

**Authors' response:** Thank you. Indeed, it is fairer by using multi-criteria to do both the calibration and assessment. However, in our study, we aim to identify the relationship between forecast skill and forecast-driven reservoir operation. To answer this question, five input combination scenarios are investigated and two of them are then applied to drive the multi-objective reservoir operation optimization. Accordingly, we prefer to distinguish the forecast skill of different configurations using the indictors of NSE, RMSE, and MAE, rather than improving the forecast skill. But it may be interesting to obtain forecasts when accounting for multi-criteria over both calibration and assessment period. We will add some discussion in the Part "Limitation and future work". Please see 516-519, Page 21.

*"Our work suffers from some limitations, which could be overcome in future studies. One of the limitations is that the single indictor is used to calibrate the forecast models while multiple indictors are used in assessing the performance of the models. It should be a fairer practice by using multi-criteria to do both the calibration and*

*assessment and can be interesting as a future work.*"

**Comment #7:** Please justify why the Radial Basis Functions are used to parameterize the policy (Line 199)?

**Authors' response:** Thank you for your suggestion. We have modified it in the new version. Please see Lines 203-206, Page 9.

*"Different DPS approaches have been proposed, where two nonlinear approximating networks, namely artificial neural networks (ANNs) and radial basis functions (RBFs) have become widely adopted as universal approximators in many applications (Deisenroth et al., 2013; Giuliani et al., 2016). In particular, we parameterize the operating policy as RBFs, because they have been demonstrated to be effective in solving multi-objective water resources management problems (Giuliani et al., 2014; 2015)."*

**Technical Corrections:**

**Comment #1:.** Please rewrite the term $(\sum_{i=1}^{K} w_k f_k)$ in equation (19), which just does not make sense to me, with the $f_k$ being a model.

**Authors' response:** Thanks for your comments. We have revised "model $f_k$" as "model forecast $f_k$.".

**Comment #2:** Please double check all the mathematical expressions.

● In equations (19) and (20), the sum should be operated over subscript "$k$" rather than "$i$".

**Authors' response:** Thank you. We have changed "$i$" to "$k$".

$$E\big[Q\big|D\big]=\sum_{i=1}^{K} w_k \cdot E\Big[ p_k\big(Q\big|f_k,D\big)\Big]=\sum_{i=1}^{K} w_k f_k \quad , \tag{19}$$

$$V\big[Q\big|D\big]=\sum_{i=1}^{K} w_k \cdot \Bigg[ f_k - \sum_{i=1}^{K} w_k f_k \Bigg]^2 + \sum_{i=1}^{K} w_k \sigma_k^2 \quad , \tag{20}$$

● It might not be right that the subscript "k" on the left side does not appear on the right side of the equation (28)

**Authors' response:** Thank you. We have revised it in the new manuscript as bellows.

$$\varphi_{i,k}(\Gamma_t)=\exp\Bigg[ -\sum_{j=1}^{M} \frac{\Big[ (\Gamma_t)_j - c_{j,i,k} \Big]^2}{b_{j,i,k}^2} \Bigg], \tag{33}$$

● It sounds not right to me in equation (34), where a variable without subscript "$j$" is summed over "$j$".

● It is questionable that the equation (35) does not have a subscript for the first sum operator to operate over.

● Expressing a variable subscript "$n$" in "$Q_{max\_n}$" (Line 247) is something strange.

● Please check on all similar unprofessional expressions in (39), (42) and (43).

**Authors' response:** Thank you. We have modified these equations. Please see Lines

251-282, Page 11-13.

"

[revised manuscript text omitted]

(2) Reservoir storage limits: $\qquad V_{t,i,\min} \leq V_{t,i} \leq V_{t,i,\max}$, $\qquad$ (47)

(3) Reservoir release limits (for the reservoir that supply water without pump station): $\qquad Q_{t,i}^r \leq Q_{t,i,\max}^r$, $\qquad$ (48)

(4) Pumping station limits: $\qquad Q_{t,j}^p \leq Q_{\max,j}^p$, $\qquad$ (49)

where $I_{t,i}$ is the inflow of the $i^{th}$ reservoir at $t^{th}$ time step (m³/s); $V_{t,i}$ is the storage of $i^{th}$ reservoir at $t^{th}$ time step (m³); $V_{min}$ and $V_{max}$ are the lower and upper storage boundaries, respectively (m³); $Q_{t,i,\max}^r$ is the maximum release of the $i^{th}$ reservoir at $t^{th}$ time step (m³/s). In some cases, $Q_{t,j}^p$ obtained by the RBF policies can be greater than $Q_{j,\max}^p$, and we will do the following step $Q_{t,n,j}^{r,2}{}' = \dfrac{Q_{t,n,j}^{r,2}}{\sum_{n=1}^{N_j} Q_{t,n,j}^{r,2}} \times Q_{j,\max}^p$ to update $Q_{t,n,j}^{r,2}$."

**Comment #3:** Please do not omit subscripts in mathematical symbols. And for all the definitions of math symbols, all the subscripts in any symbol should appear in its definition.

**Authors' response:** Thank you. We have double checked all the subscripts in mathematical symbols.

**Reply to the comments from Referee #2,**

**Comment #1:** deterministic S5 performance indicators overlap each other, I would suggest to modify color or width to improve the readability.

**Authors' response:** Thanks for your constructive comments. We have revised this figure in the new version.

[Figure]

**Figure 6: NSE values at lead times of 1 to 7 days plotted against the coefficient of variation (COV) for all the 24 reservoirs during the period of (a) calibration, (b) validation, and (c) test under S5.**

---

## Referee Report (RR1)

**HESS-2020-617R1**

The authors have made very good efforts in clarifying my concerns. This work aims to accomplish tasks in dealing with a real-world problem, which should be encouraged. The manuscript, however, is not well organized and well written, making it very difficult to catch the logical relationship among quite many models and methods.

**Specific Comments:**

(1) It is very important that the Figure 1, in which the MOEAs and NSGA-II should appear, must be logically and carefully explained.

(2) It should be clearly explained on how the hydrological variables are related to the mathematical variables in LSTM, RGU and GWO-LSSVM. By the way, the model forecast ($f$) in (18) cannot be found in the model RGU, why?

(3) Why you need generate a realization of observation ($y_t$), rather than using the historically observed (Line 177)? Also, it is not clear where this "$y_t$" has been used in the following models.

(4) Based on the procedure in Lines 175-179, why you need simulate a large number (M =1000) of forecasts since there are only three optional forecasts to be repeatedly chosen from?

(5) From Line 214 to 225: this is where you should make very good efforts to logically clarify how the MORDM is coupled with the deterministic and stochastic forecasts, the DPS, the optimization problem of up to 7 days ahead with the MOEAs /NSGA-II, especially, how the performances /objectives are evaluated when optimizing the operating policy ($p_\theta$)?

(6) Please check on Line 281, why it is the release from a reservoir that is refined to make a pumping flow feasible?

(7) Please improve on the optimization problem formulation in a more professionally mathematical way. The relationship between variables is incomplete. The hydraulic relationship / connections, for instance, between reservoirs, pumps, channels and water demanders, are not logically presented.

**Technical Corrections:**

(1) The "g" in (21) is left unexplained.

(2) In Line 208, "$\Gamma_t$ is the $i^{th}$ policy inputs", should it be "$t^{th}$"?

(3) In (33), $(\Gamma_t)_j$ is left not denoted;

(4) The equation in Line 282 should be indexed.

(5) Please replace the typo "fjor" in Line 520 with "for".

(6) The typos in (19) and (20) still remain unchanged. The sum operator should be over subscript "$k$" instead of "$i$".

---

## Author Response (AR2)

Dear Referee #1,

We highly appreciate your review and useful comments on our manuscript again. We have made appropriate revisions according to your comments and provided our responses to your queries below.

Kind regards, all authors

Queries by anonymous referee #1 & answers by authors are as follows:

**Specific Comments:**

(1) It is very important that the Figure 1, in which the MOEAs and NSGA-II should appear, must be logically and carefully explained.

**Authors' response:** Thanks for your constructive comments. The multi-objective evolutionary algorithms (MOEAs) are used to optimize the parameterized policies. The MOEAs associated with the AI-based management methodology is present in Figure 1.

**Figure 1: Framework of the AI-based management methodology.**

We have added more explanations in the revised version to explain how MOEAs are used in the parameterized MORDM method and presented in Figure 3. Please see Page 11, Lines 243-246.

"(4) Optimizing the parameterized policies using multi-objective evolutionary algorithms (MOEAs) based on the robust performance objectives. Repeat Steps (2), (3), and (4) until the times of population iteration are reached and then export the optimal Pareto solutions. In this study, the optimization is solved by applying NSGA-II to search the space of decision variables and identify the trajectories.

---

## Author Response (AR3)

Dear Referee #1,

We highly appreciate your review and useful comments on our manuscript again. We have made appropriate revisions according to your comments and provided our responses to your queries below.

Kind regards, all authors

Queries by anonymous referee #1 & answers by authors are as follows:

**Specific Comments:**

(1) The authors have made very good efforts to address my concerns, and the manuscript is more readable. A typo only: please replace the "subjecting to " with "subject to" (Line 228)..

**Authors' response:** Thanks for your comments. We are sorry for it and have replaced the "subjecting to " with "subject to. Please see Page 10, Line 228.

*"(2) Generate alternative parameterized policies subject to all the constraints, and the objectives are evaluated over stochastic inflows with the following procedures (Giuliani, et al., 2016)"*